



# Positive and negative human-modified droughts: a quantitative approach illustrated with two Iranian catchments

Elham Kakaei[1,3], Hamid Reza Moradi[1], Ali Reza Moghaddam Nia[2], Henny A.J. Van Lanen[3]

[1]Department of Watershed Management Engineering, College of Natural Resources and Marine Sciences, Tarbiat Modares
University, Noor, 46414-356, Iran
[2]Department of Range and Watershed Management, Faculty of Natural Resources, University of Tehran, Karaj, 1417466191, Iran
[3]Hydrology and Quantitative Water Management Group, Wageningen University, Wageningen, 6700 AA, the Netherlands

*Correspondence to*: Hamid Reza Moradi (hrmoradi@modares.ac.ir)

**Abstract.** Drought management in the Anthropocene is challenging because of the complex interaction between climate variability, land surface processes and human changes in the hydrologic system. In the current research, different kinds of natural and human-affected droughts in the Anthropocene, including climate-induced drought, human-induced drought and human-modified drought have been investigated for two Iranian catchments (Kiakola and Eskandari catchments), as an example. We propose a methodology to distinguish and quantify positive and negative human-modified droughts to explore
the impact of human interferences on river flow and groundwater. The methodology uses naturalized conditions obtained by simulation modeling as a reference to distinguish the droughts. Positive human-modified droughts happen when human activities alleviate natural droughts, whereas negative human-modified droughts reflect severer drought conditions than naturally occurring. Application of the methodology shows that human activities mostly caused negative human-modified droughts (e.g. decreased river flow in more than 89% of the events) in the selected Iranian catchments, where the Eskandari
catchment had about four times severer human-modified droughts than the Kiakola catchment. Positive human-modified droughts mostly occurred in the wet season, which cannot counterbalance the adverse impacts of the negative human-modified droughts occurring in the following dry season. The proposed methodology enables further quantification of drought in the Anthropocene through evaluation of negative and positive effects of human activates on the hydrologic system, which will support water management to reduce drought impacts.

**1 Introduction**

Drought is a recurrent natural hazard, which affects all parts of the hydrological cycle and can lead to drastic impacts all around the world (Wilhite, 2000; Wilhite, 2005; Tallaksen & Van Lanen, 2004; Mishra & Singh, 2010; Sheffield & Wood, 2011; Vicente-Serrano et al., 2014; Stahl et al., 2016). According to the occurrence of drought in different parts of the hydrological cycle, drought is categorized into three types, including meteorological drought, soil moisture drought, and
hydrological drought, that is, drought in groundwater and streamflow (Wilhite & Glanz, 1985; Tallaksen & Van Lanen,



2004; Sheffield & Wood, 2011). The severe negative impacts of drought on different sectors (e.g. environment, , agriculture, energy) cause it to be a major concern, particularly under global warming (Zhao & Running 2010; Prudhomme et al., 2014; Wanders & Wada, 2015; Chen & Li., 2016). Natural hydro-climate variability and human interventions (e.g. irrigation, reservoir operation, exploitation of groundwater) are the main reasons for occurrence of hydrological drought, which is

relevant for water resources management (Van Loon et al., 2016a; Yuan et al., 2017). Drought management in the human-influenced era, which considers the multi-directional relationship between natural hydro-climate variability (natural drought processes) and human interventions (societal processes), plays a vital role to minimize drought impacts, and lack of knowledge about this complex relationship impedes drought management (Van Loon et al, 2016a; 2016b). Human activities affect the environment both directly (e.g. water abstraction) and indirectly (e.g. climate change associated with greenhouse

gas emissions) (Wagener et al., 2010). Direct modification of drought severity by human-influences are most recognized and understood (Van Loon et al., 2016a).

So, for better understanding of interactions between climate, hydrology and humans, the effect of anthropogenic activities on drought needs to be assessed (Rangecroft et al., 2016). In this context, drought caused solely by natural drivers or human processes is defined as "climate-induced" or "human-induced", respectively (Van Loon et al, 2016a). In addition, as a result

of anthropogenic drivers of drought, "human-modified drought" has been introduced, which reflects drought conditions caused by an interplay of natural and societal processes. This distinction in drought addresses different causes. Human-modified drought is anticipated to be a common phenomenon in the arid and semi-arid regions of the world, which typically are low in total annual precipitation (Yazdandoost, 2016). Increasing pressures on water resources in these dry regions have led to increased water use, incl. the storage of water in dams, large-scale irrigation. Iran is a distinctive example of a water-

short country with an average rainfall of 250 mm/year (Motiee et al., 2012). Last decades, drought has been a recurrent phenomenon in Iran. These factors contribute to the water crisis there, which finds its basis in: (i) rapid population growth, which is disproportionate to the environmental capacity, (ii) development of agriculture, industry and cities, (iii) a decrease in the number of appropriate structures to store, distribute and convey water (this is due to the lack of financial sources, which has led to the insufficient investments), and (iv) more frequent occurrence of drought (Motiee et al., 2001).

More knowledge is required to combat the water crises in dry environments, like in Iran. In this context a further elaboration of human-modified drought is crucial. Focus usually is on negative impacts of (human-modified) drought (e.g. Stahl et al., 2016), which, clearly are important. However, to obtain the full overview, positive effects of human-modified drought also required to be considered, that is, some human interferences are meant to alleviate impacts of climate-induced drought. This paper aims to propose a step-wise methodology to quantify the distinction between different drought types in the

Anthropocene, in particular making a distinction between positive human-modified (alleviation of natural drought) and negative human-modified droughts (enhancement of natural drought). The methodology is explained by using two case studies in Iran with arid and semi-arid climate conditions, where human-modified droughts are a major concern.

The paper is organized as follows. First, in Section 2.1, the study areas in Iran are described, incl. the data. Next, the methodology is explained that identifies and quantifies positive human-modified and negative human-modified droughts



among the other drought types (Sections 2.2, 2.3, 2.4, and 2.5). In Section 3, quantitative results of natural and human-affected drought types with focus on positive and negative human-modified droughts are presented, which is followed by Section 4, in which the research is discussed and conclusions are drawn (Section 5).

## 2 Materials and methods

5 The methodology to distinguish between human-modified droughts is presented in Figure 1. Here we present a brief description about each step. Step 5 is new and an addition to the methodology introduced by Van Loon and Van Lanen (2013).

### 2.1 Study areas

### 2.1.1 Eskandari catchment

10 The Eskandari catchment is located west of Esfahan city, in the northern part of the Zayandehrood dam basin, Iran (Figure. 2, left). The catchment (area about 1649 km$^2$) is located between the longitudes of 50° 2′ to 50° 41′ and the latitudes of 32° 12′ to 32° 46′ and its elevation varies between 2116 to 3877 m.a.s.l (Eslami et. al. 2017). The most important river is the Pelasjan River and the majority of the catchment consists of flat lands with a climate ranging from very humid in the upstream parts of the catchment to semi-humid near the outlet. According to the Köppen-Geiger climate classification 15 (Kottek et.al. 2006) the dominate climate is BSk (main climate: arid; precipitation: steppe; temperature: cold arid). The average annual precipitation is about 420 mm, which falls mostly in fall, winter and early spring (November-March). In this catchment most of the precipitation takes place as rain and there is some snow in the mountains. The catchment's runoff is measured at the Eskandari hydrometric station. In this study, we used three rain gauge stations (Eskandari, Badijan and Damaneh-Fereidan); one is the Badijan climatology station. There are 22 groundwater wells in the Eskandari catchment that 20 are well distributed and have long time series. The selected period for modelling was 1976-2014, for which time series of hydrological and meteorological data were available. The groundwater data were available for the period 1983-2014.

### 2.1.2 Kiakola catchment

The Kiakola catchment with an area of about 2100.9 km$^2$ is located in the central section of the Alborz Mountains, in the north of Iran (Figure 2, right). The precipitation is rain in most part and snow in mountainous part. The Kiakola catchment is 25 located within 35°44' to 36°19' latitudes and 52°35' to 53°23' longitudes and the elevation varies between 3991 to 9 m.a.s.l. The average annual precipitation is 700 mm (Gholami et. al. 2016) and the climate in this catchment also is categorized as BSk (Kottek et.al. 2006). In this study, we used three rain gauging stations (Kiakola, Alasht and Chashem) and one hydrometric station (Kiakola in Shirgah). There was no representative groundwater well in this catchment. So, no



groundwater drought analysis could be done. The selected period for modelling was 1976-2013, for which time series of hydrological and meteorological data were available.

## 2.2 Subdivision of the period in natural and disturbed (Step 1)

Assessment the effect of anthropogenic activities on drought leads to provide valuable information on interactions between
climate and humans activities. In research on drought types, long time series of hydrological and meteorological data are required that include natural and disturbed conditions. Through statistical analyses, time series of hydrological and meteorological variables can be divided into a natural period and a disturbed period (Step 1, Figure 1). Based on differences in time series from these periods, the effects of climate variability and human activities on hydrological variables can be distinguished. Trend tests can be used to explore if different trends can be observed in meteorological variables and
hydrological variables, which might point at human impact on, for instance, river flow or groundwater levels. In this study the Mann-Kendall trend test (Mann, 1945; Kendall, 1975; Yue & Wang, 2004; Wang et al., 2013) was applied.

To identify change-points in a time series of a hydrological and meteorological variable other statistical tests can be used. In this study the Pettitt's test (Pettitt, 1979) was used to determine the occurrence of a change-point in the Iranian time series. Then, we calculated the probability of change-point for each year by the formula proposed by Kiely et al. (1998).

### 2.2.1 Mann-Kendall Test

In order to detect trends in hydro-meteorological time series, the Mann-Kendall test was utilized. In this test, the null hypothesis, $H_0$, implies that the data $(x_1, x_2, ..., x_n)$ are a sample of $n$ identically and independent disturbed random variables. While the $H_1$ indicates that the two-sided test of distribution of $x_k$ and $x_j$ for all $k, j$ are not identical (Li et al., 2016).

$$S = \sum_{k=1}^{n-1} \sum_{j=k+1}^{n} sgn(x_j - x_k) \,,$$                                   [Equation 1]

Where $x_k$ and $x_j$ are from $k = 1,2, ..., n-1$ and $j = k+1, ..., n$, respectively. Also,

$$sgn(\theta) = \begin{cases} 1 & \theta > 0 \\ 0 & \theta = 0 \\ -1 & \theta < 0 \end{cases}$$                                   [Equation 2]

In addition, $Z_c$ and $\beta$ parameters are given as following:

$$Z_c = \begin{cases} S - 1/\sqrt{var(S)} & S > 0 \\ 0 & S = 0 \\ S + 1/\sqrt{var(s)} & S < 0 \end{cases}$$                                   [Equation 3]

Where $Z_c$ is the test statistic. , $H_0$ will be rejected when $|Z_c| > Z_{1-\propto/2}$, in which $\propto$ and $Z_{1-\propto/2}$ are the significance level and standard normal deviation, respectively. The trend magnitude is given as following:





$$\beta = Median\left(\frac{x_j - x_k}{j-k}\right), \forall k < j \qquad\qquad \text{[Equation 4]}$$

Where $1 < k < j < n$. Positive and negative value of $\beta$ indicates an upward and downward trend, respectively.

Mann-Kendal trend tests were done (Annex 1) on the time series of the hydrometeorological variables. In the Eskandari catchment, the discharge and temperature showed clear significant downward and upward trends, respectively. While, rainfall and evapotranspiration showed a weak none-significant upward trend. In the Kiakola catchment, discharge showed a significant downward trend and temperature and rainfall showed significant upward trends. In addition evapotranspiration showed none-significant upward trend. The results indicate the influence of both natural drivers (weather variables) and human activities (discharge), although the higher potential evapotranspiration due to the raised temperature could also have to some extend effect on the river flow.

### 2.2.2 Pettitt's Test

In order to determining the occurrence of a change point and make a distinction between natural period (undisturbed period) and disturbed period, the non-parametric approach which developed by Pettitt (1979) was utilized. When the exact time of change is unknown, the Pettitt's test detects for a significant change in the mean of time series. Pettitt's test analysis the $H_0$, as the null hypothesis, when the $T$ variables follow one or more distribution that have the same location parameter (absence of a changing point). The non-parametric statistic is defined as following (Ma et al., 2008):

$$K_T = max|U_{t,T}| \qquad\qquad \text{[Equation 5]}$$

where

$$U_{t,T} = \sum_{i=1}^{t} \sum_{j=t+1}^{T} sgn(x_i - x_j) \qquad\qquad \text{[Equation 6]}$$

The change point is located at $K_T$ and the statistic is significant. The significance probability of $K_T$ is approximated for $p \leq 0.05$ with

$$p \cong 2exp\left(\frac{-6K_T^2}{T^3 + T^2}\right) \qquad\qquad \text{[Equation 7]}$$

The Pettitt's test was done on the discharge time series for both catchments to find change-points that enabled us to distinguish between periods without and with clear human influences (Annex 2). According to the test the period 1976-1996 could be classified as the natural period (undisturbed period) and the period 1996-2014 as disturbed in the Eskandari catchment. For the Kiakola catchment the periods 1976-1998 and 1998-2013 could be considered as natural and disturbed, respectively (Annex 2).



**2.3 Naturalizing time series of hydrological variables through hydrological modelling (Step 2)**

Due to intensive human activities in the case studies which influenced the river flow (e.g. abstraction of river water for irrigation purposes) and lack of long time series of state variable (groundwater storage) hydrological modeling is needed. In current research, hydrological modeling was done to achieve two main purpose: (i) naturalized the disturbed river flow time
series to identify human effects (Step 2, Figure 1), and (ii) simulation of ground water storage for which no long time series exist. Several approaches are available (Rangecroft et al., 2016), among others hydrological modelling (e.g. Van Loon and Van Lanen, 2013; 2015).

**2.3.1 Hydrological modeling by HBV**

In this study the HBV (Hydrologiska Byråns Vattenbalansavdelning) hydrological model, light version, was applied due to
three main reasons: (i) simple and flexible model structure, (ii) HBV is not intensive data model and input data required were available, and (iii) the model has been widely utilized in different climate condition also in semi-arid environment (e.g. Masih et.al., 2010; Love et al., 2010; Lidén and Harlin, 2000). HBV as a conceptual, semi-distributed, rainfall-runoff model, which is developed by the Swedish Meteorological and Hydrological Institute (SMHI), simulates daily discharge from daily precipitation and temperature, and monthly or daily estimates of potential evaporation. The model consists of different
routines representing the snow accumulation and snowmelt are calculated by a degree–day method, recharge and actual evapotranspiration as functions of the actual water storage in a soil box, runoff generation by two linear reservoirs in series with three possible outlets (i.e., runoff components) in the STANDARD respond routine, and channel routing which is computed by a simple triangular weighting function. In addition, the DELAY response routine has two parallel linear reservoir. Lower reservoir is preceded by a distribution of recharge over different delay boxes. According to the Seibert
(2000, 2005), for hydrological modelling of slowly responding deep-groundwater catchments the DELAY response routine is better than STANDARD response routine. Further description of the model can be found in Seibert (1999, 2000, 2005), Seibert et al. (2003), and Seibert and Vis (2012).

**2.3.2 HBV calibration and validation**

For Eskandari and Kiakola catchments, the HBV model by a daily time step was used with observed meteorological data to
illustrate the methodology. For each catchment, precipitation data from different stations averaged using Thiessen polygons and potential evaporation was calculated using FAO Penman-Monteith (Allen et al., 1998). In order to hydrological modeling by HBV, the first four (1976-80) and seven (1976-83) recorded years were used to warm-up period prior to estimate the initial state parameters since of the significant inter-annual climate variability in Eskandari and Kiakola catchments, respectively. For Eskandari catchment, the rest of natural (undisturbed) period (1981-96) was divided in to the
two period including (1981-90) for calibration and (1990-96) for validation. In addition, for Kiakola catchment, the period of (1984-92) and (1992-98) was considered as calibration and validation period, respectively. The discharge of the disturbed



period could be naturalized by calibrating the HBV model with undisturbed (natural) period and applying the calibrated model to disturbed period. The DELAY response routine was used and values of HBV parameters were determined by the genetic calibration algorithm which is described by Seibert (2000). The main focus of calibration was on the correctly reproducing observed discharge, especially on low flow discharge. So, Nash-Sutcliffe efficiency according to the algorithm

of the observed and simulated flow discharge ($lnReff$), as the best objective function for low-flow modelling, was utilized to assessment between observed and simulated discharge. The calibration procedure was done more than fifty times, based on visual inspection and $lnReff$ efficiency, and the best result was selected for further steps. Because of the random elements of the Genetic Algorithm and Powell optimization (GAP optimization) used for calibration (Vis et al.,2015), each calibration was repeated 100 times as calibration trials, which resulted in 100 different parameterizations. The structure of

HBV model, by DELAY routine response, involved 14 parameters is shown in Annex 3 (Table 1).

The results of HBV modelling for the calibration and validation periods are shown in Table 1. For the calibration period, the $lnReff$ and $R^2$ in both catchments were greater than 0.50 and 0.60, respectively, which indicates an acceptable performance of the HBV model for both catchments. Since of much more irregular seasonal variation in Eskandari catchment, $lnReff$ value for Eskandari is lower than the value for Kiakola catchment. For the verification period, the $lnReff$ and $R^2$ are

slightly lower, but reflect the capability of HBV to model the runoff process in a wide range of environments (e.g. Seibert and Vis, 2012). Figure 4 shows the observed and simulated daily discharge time series for both catchments. The simulated discharge agrees reasonably well with the observed for the calibration period. In the calibration period HBV underestimates some peak discharges, which resulted in a lower $lnReff$. Overall, the observed discharge was slightly more peaky than the simulated discharge. However, seasonal and inter-annual variability in discharge have been acceptably simulated by HBV.

Comparison between annual and monthly values of the $50^{th}$ and $80^{th}$ percentile of the duration curves of observed and simulated discharge is shown in Annex 3 (Table 2). As can be seen from the percentiles, in both catchment simulated discharge percentiles showed reasonable agreement to observed percentiles and inter-annual variation was reproduced well. So, simulated discharge can be used as an approximation of the natural discharge to quantify climate-induced drought, human-induced drought and human-modified drought in the study regions. Simulated groundwater for the Eskandari

catchment was evaluated as well. The $R^2$ of 0.4 is rather low but yet acceptable knowing that observed groundwater levels are site specific, which had to be derived from spatially-lumped groundwater storage simulated with HBV applying a constant storage coefficient. In general, the acceptable performance of the HBV in the natural period for both catchments indicated the ability of the model to simulate the naturalized hydrological situation in the disturbed period.

## 2.4 Applying a threshold

The drought analysis in the proposed methodology (Step 3, Figure 1) was based upon the threshold level approach (Yevjevich, 1967; Hisdal et al., 2004). When the hydrometeorological variable of interest (e.g. river flow) is below a





predefined threshold a drought occurs. A drought event starts when the variable falls below the threshold level and the event continues until the threshold is exceeded again. A variable threshold often is used, as seasonal patterns are then taken into account. It addresses deficiencies in the wet season that can lead to a drought in the subsequent dry season (Hisdal and Tallaksen, 2000), and it also detects the actual droughts in the dry season. For a time series of variable $X_i = \{x_1, x_2, ..., x_n\}$,

in which i is the kind of variable (i.e. discharge), a drought occurs when the variable of interest $(X_{i,t})$ is equal or below a predefined threshold $(\tau_i)$ (onset; t=1) and the event continues until the threshold is exceeded again (recovery; t = T).

For both catchments, the threshold values calculated for natural (undisturbed) period and applied to the entire time series. By applying a centred moving average of 30 days, the monthly threshold were smoothed. Mutually dependent drought were pooled using inter-event method and based on the range given by Tallaksen et al. (1997) and Fleig et al. (2006), time period

of 10 days was used for both catchments. In order to eliminate minor droughts, all droughts events with duration less than 15 days were omitted from the analysis.

### 2.5 Anomaly analysis

According to the definition of a climate-induced drought (caused by only climate variability, Van Loon et al., 2016a; 2016b), such an event occurs when the naturalized time series of the variable of interest $(X_{[N_{i,t}]})$ is below the threshold $(\tau_{i,t})$.

Similarly to Tallaksen et al. (2009) it is defined as:

$$Climate - induced\ drought: \delta_{i.t} = \begin{cases} 1 & X_{[N_{i,t}]} - \tau_{i,t} \leq 0 \\ 0 & X_{[N_{i,t}]} - \tau_{i,t} > 0 \end{cases},$$   [Equation 8]

In which $(\delta_{i,t})$ is a binary variable indicating the drought situation on time t. The threshold $(\tau_{i,t})$ is derived from the time series of the natural conditions.

In the absence of natural drivers of drought $(X_{[N_{i,t}]} - \tau_{i,t} > 0)$, a human-induced drought (drought caused only by human

influence) may occur when the observed time series of the variable of interest $X_{[obs_{i,t}]}$ is below the threshold (Step 4, anomaly analysis 1, Figure 1):

$$Human - induced\ drought: \delta_{i.t} = \begin{cases} 1 & X_{[N_{i,t}]} - \tau_{i,t} > 0\ and\ X_{[obs_{i,t}]} - \tau_{i,t} \leq 0 \\ 0 & X_{[N_{i,t}]} - \tau_{i,t} > 0\ and\ X_{[N_{i,t}]} - \tau_{i,t} > 0 \end{cases},$$   [Equation 9]

A human-modified drought, which is caused by combination of climate variability and human influence, is defined as an anomaly in the observed time series $(X_{[obs_{i,t}]})$ and the naturalized time series $(X_{[N_{i,t}]})$ of the variable of interest relative to the

threshold $\tau_{i,t}$ at time t (Step 4, anomaly analysis 1, Figure 1). It is defined as follows:

$$Human - modified\ drought: \delta_{i.t} = \{1\ X_{[N_{i,t}]} - \tau_{i,t} \leq 0\ and\ X_{[obs_{i,t}]} - \tau_{i,t} \leq 0\ ,$$   [Equation 10]





Positive and negative human-modified droughts are estimated by comparing $X_{[obs_{i,t}]}$ and $X_{[N_{i,t}]}$ as follows (Step 5, anomaly analysis 2, Figure 1):

$$X_{[N_{i,t}]} - \tau_{i,t} \leq 0 \ and \ X_{[obs_{i,t}]} - \tau_{i,t} \leq 0 \ if \begin{cases} X_{[obs_{i,t}]} < X_{[N_{i,t}]} & Negative \\ X_{[obs_{i,t}]} \geq X_{[N_{i,t}]} & Positive \end{cases},$$ [Equation 11]

A climate-induced drought that is enhanced by human drivers is called a negative human-modified drought, whereas a

5 positive human-modified drought reflects conditions that human interventions alleviate the climate-induced drought. A schematic climate-induced drought, a human-induced drought and positive and negative human-modified droughts are demonstrated in Figure 3. Matlab programing, Matlab version R2014a, utilized to make a distinction between climate-induced drought, human-induced drought and human-modified drought.

The duration of the different drought types is defined as the number of uninterrupted time steps with a variable of interest

below the threshold, and the average deficit volume as the sum of the deficit volume over an uninterrupted number of time steps with a variable below the threshold (Tallaksen et al., 2009).

By applying the described methodology, different kinds of threshold can be defined. We applied a monthly threshold derived from the 80[th] percentile of the monthly duration curves (Van Loon et al., 2010). The choice of a different percentile in the calculation of the threshold level affects the magnitude of the drought characteristics (Van Loon and Van Lanen, 2012).

**3 Results**

Here, we present the outcome of the stepwise application of the methodology to quantify positive and negative human-modified droughts (Figure 1) using the Eskandari and Kiakola catchments in Iran as examples.

**3.1 Distinction between climate-induced drought, human-induced drought and human-modified drought (Steps 3 and 4)**

First, a threshold should be determined to quantify climate-induced drought, human-induced drought and human-modified drought. A variable threshold has been applied in both catchments to represent the seasonal variability. We applied a monthly threshold derived from the 80[th] percentile of the monthly discharge series. The threshold values were calculated based on the observed discharge time series from the natural period (Van Loon & Van Lanen, 2012). So, the calibrated HBV model (which is calibrated for natural period) was run for disturbed period to get naturalized flow and groundwater time

series. Then, the natural drought and the two human-affected drought types can be distinguished by comparing the naturalized, observed and threshold discharge and groundwater levels, as proposed by Van Loon and Van Lanen (2013; 2015) (Section 2.2).

Climate-induced droughts (solely caused by climate variability) were identified by connecting the naturalized discharge time series with the monthly-varying discharge threshold (Eq 8). Tables 2 and 3 provide some summary statistics of the climate-





induced droughts for the two Iranian catchments and Figure 5 shows through the hydrographs the occurrence of the droughts. In the study period 41 and 34 events occurred in the Eskandari and Kiakola catchments, respectively. In the Eskandari catchment, the mean and maximum water deficit and mean duration of the climate-induced droughts are higher than in the Kiakola catchment, which is mainly because of the lower climate variability in the dryer Eskandari catchment.

Human-induced droughts (solely caused by human activities) were determined by comparing the observed discharge time series and monthly-varying threshold (Eq 9). Several severe human-induced droughts occurred in both catchments (Figure 6). Like, for the climate-induced drought, the number of this type of drought in the Eskandari catchment is higher than in the Kiakola. In addition, this type of drought last longer in the Eskandari catchment and is severer than in the Kiakola catchment, that is, the mean duration is about twice as long and mean deficit more than 3 times larger (Tables 2 and 3)

Human-modified drought, which are caused by a combination of climate variability and human influence, were derived from comparing the time series of observed and natural discharge, and the monthly-varying threshold (Eq 10). The number of human-modified droughts in both catchments are about equal but, the Eskandari catchment suffered from longer and more severe droughts. The human-modified droughts are more than 2.6 times longer and more than 3.7 times severer than in the Kiakola catchment. Figure 7 shows the human-modified droughts at the Eskandari and Kiakola gauging stations.

Table 2 shows that the number of human-modified droughts is almost equal to the number of climate-induced droughts, and lower than the human-induced droughts in the Eskandari catchment. However, the mean and maximum duration and deficit volume of human-modified droughts was substantially larger than of the other two types. In the Kiakola catchment the number of human-modified droughts is higher than climate-induced and human-induced droughts (Table 3). However, the mean and maximum duration and deficit volume of the human-modified drought is not always the largest. Further examples

of climate-induced drought, human-induced drought and human modified drought are shown in Annex 4.

Characteristics of drought in groundwater in the Eskandari catchments are shown in Table 4. In the study period, there were less groundwater droughts than droughts in river flow (Table 2). Striking is the length of the human-modified groundwater drought (196 months), which lasted for more than 16 years. This point at long-lasting overexploitation of groundwater in the Eskandari catchment. During these years the groundwater has failed to recover from precipitation, which indicates the large

human influence on groundwater storage.

### 3.2 Distinction between positive and negative and human-modified drought (Step 5)

As a last step in the newly proposed methodology (Figure 1), a distinction has been made between positive and negative human-modified droughts through an anomaly analysis (Eq 11). The distinction enables evaluation of alleviating or enhancing effects of human drought drivers. In both catchments substantially more events were classified as negative

human-modified droughts (Tables 3 and 4), that is, an increase of negative pressures on the hydrological system due to human activities. Although the climatic conditions of these two Iranian regions are rather different, the negative pressure on the system in terms of droughts is almost the same. In the Eskandari catchment, 96.9% of events were categorized as negative human-modified droughts and clearly the rest were considered as positive-human-modified droughts. In the Kiakola



catchment, 88.9% of human-modified drought events were classified as negative human-modified drought. In both catchments, negative human-modified droughts are longer and more severe than positive human-modified droughts. Clearly, in groundwater (Eq 11), all drought events have been classified as a negative human-modified drought (Table 4). In Figure 8, the combined effect of natural and human drivers of drought in groundwater storage is shown.

**4 Discussion**

In this study we propose a methodology to identify negative and positive human-modified droughts (Figure 1, Eq 11), which extends the approach introduced by Van Loon and Van Lanen (2013; 2015) and Van Loon et al. (2016a). Two Iranian catchments with different climate conditions were used to illustrate the step-wise methodology. As an alternative to the simulated/observed approach (this study), human influence on hydrological variables in the methodology (Step 2) could be
investigated through (i) paired catchment analysis, or (ii) upstream/downstream analysis (Rangecroft et al., 2016). Monthly-variable thresholds, that is, river flow and groundwater that is exceeded or equalled in 80% of the time, were determined (Step 3) for the drought identification. We explored the use of other thresholds for drought identification through the anomaly analyses (Step 4). We tested the effect on drought characteristics, and eventually on the percentage of negative and positive human-modified droughts (Step 5) by using the $50^{th}$, $70^{th}$ and $90^{th}$ percentiles. Table 5 provides the drought
characteristics of the human-modified for the different defined thresholds, which were derived from 50th, $70^{th}$ and $90^{th}$ percentile of the discharge time series. As hypothesized, choosing different percentiles as threshold will change the magnitude of the drought characteristics. A lower threshold ($90^{th}$ percentile) leads to fewer human-modified events with shorter durations and lower deficits, whereas, higher thresholds ($50^{th}$ and $70^{th}$ percentiles) results in the opposite. However, more importantly, the percentage of negative and positive human-modified does not change substantially. The percentage
negative human-modified droughts remain high, i.e. around 80-95%, which indicates that this is not too sensitive to the selected threshold. Table 5 shows that choosing a different percentile for the threshold also will slightly change the portion between negative and positive human-modified groundwater droughts, that is, positive human-groundwater drought will vary between 0 to 18%.

The results show that most of the time human activities have a negative influence on the discharge in the selected Iranian
catchments (negative human-modified drought). In the Eskandari catchment wet months occurred in the period November-March. As shown in Figure 9, positive human-modified droughts mostly occurred in the wet months, whereas the substantially larger deficits in discharge (negative human-modified droughts) appeared during the dry season (summer) (Figure 10). In the Eskandari catchment the low precipitation in the dry season, increasing surface water use and the increased abstraction from wells to cover the increased water demand (Karbalaee, 2010) have led to more severe droughts
(climate-induced droughts transferred into negative human-modified droughts or even human-induced droughts, Figures 6, 7 and 8) in the dry season when the water use is largest (Tabari. et al., 2012).





Our results for the two Iranian catchment correspond well to assessment the effect of drought and human influences in the Upper-Guadiana in Spain (Custodio, 2002; Van Loon and Van Lanen, 2013) and the analysis of the "Millenium Drought" (2001–2009) in southeast Australia (Van Dijk et al., 2013). Rangecroft et al. (2016) showed that in case of groundwater abstraction, there is at least a 50% increase in deficit compared to the natural situation, which means the occurrence of

negative human-modified drought. In addition, they describe that whether negative or positive human-modified human-modified droughts occur due to dams is dependent on purpose and management of reservoirs. Moreover, it appears to be difficult to assess the sign of the human-modified droughts as a result of urbanization because divergent processes occur. Also, it is hard to determine the sign of the impact when a user captures and diverts water from a stream. How the user's consumptive use that will affect other users downstream is really complex to analyse and to quantify, and is among others

dependent on whether the water is used and (partly) put back in the stream upstream of the abstraction location (Chen & Li, 2016). These complex, interrelated processes, which may a constant source of confusion debate and conflict, act as deterrent factor for many policy makers, researches and users (Rijsberman 2006). Water management can prevent water resources overexploitation (Garrido et al., 2006) through decreasing human-induced drought and human-modified drought. Then water management only has to cope with climate variability, which causes climate-induced drought (Garrido et al., 2006, Harou et

al., 2010). A Multi-Agent Simulation approach can be applied if more information is available on the human decision making in a catchment (Van Oel et al., 2010). Such a model is often used in water management applications, but can also be used to distinguish climate-induced, human-induced and human-modified droughts (Van Loon & Van Lanen, 2015).

## 5 Conclusions

Current research proposes a step-wise methodology to expand on the distinction between different hydrological drought

types in the Anthropocene, in particular making a distinction between positive human-modified and negative human-modified droughts. A schematic outline (Figure 1) and Equation 11 are provided to make this distinction. A second anomaly analysis is introduced to investigate which percentage of the human-modified droughts is negative and which is positive, implying an enhancement or alleviation, respectively. The methodology has been illustrated using two (semi-)arid case studies in Iran. Our results indicated that hydrologic system in both catchments has faced severe negative human-modified

droughts, which lasts longer and are more severe than the droughts under natural conditions (i.e. climate-induced droughts). In the arid case (Eskandari catchment) the human-modified droughts are longer and more severe than those in the semi-arid region (Kiakola catchment).

Catchment storage and release processes strongly modify drought severity from precipitation (meteorological drought) into streamflow (hydrological drought). In the undisturbed (natural) situation, factors such as geology, land cover and soil type

are acting as modifiers (e.g. Van Lanen et al., 2013). While, in the Anthropocene land properties and storage are changed by human activities and effect propagation processes, and hence drought severity is being modified. Human modifiers can have enhancing (negative) or attenuating (positive) effects on drought duration and severity (Van Loon et al., 2016a). The



combined effects of climate-induced, human-induced and human-modified droughts have an important impact on water resources management. In this research we have made clear that drought in the Anthropocene is not an external natural hazard solely driven by climate variability, but that drought is interlocked to human influences on the water cycle. The proposed methodology, as a multi-directional and multi-driver drought framework, enables quantification drought in the Anthropocene. So, it makes it possible for water resources managers to decide on how to combat natural and human-made droughts.

**Author contribution:** EK and HVL came up with the concept for the manuscript and conducted the analysis. EK wrote the manuscript with the input from all co-authors (EK, HVL, HRM, ARMN). HVL provided continuous input and insight. HRM and ARMN provided access to the data from the both basins.

**Competing interests**: The authors declare that they have no conflict of interest.

**Acknowledgements:** The authors would like to thank the research group at the Wageningen University and Tarbiat Modares University for their cooperation. The research is supported by the ANYWHERE project (Grant Agreement No.: 700099), which is funded within EU's Horizon 2020 research and innovation program (www.anywhere-h2020.eu). This research is part of the Wageningen Institute for Environment and Climate Research (WIMEK-SENSE) and it supports the work of the UNESCO EURO -FRIEND-Water and the IAHS Panta Rhei program





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



**Table 1: Performance of the HBV model for the natural period in the two Iranian catchments**

| Catchment | Period | $R^2$ | Re $f\!f$ | ln Re $f\!f$ |
|-----------|--------|-------|-----------|--------------|
| Eskandari | Calibration (1976-1990) | 0.621 | 0.525 | 0.569 |
|           | Verification (1990-1996) | 0.616 | 0.556 | 0.501 |
| Kiakola   | Calibration (1976-1992) | 0.622 | 0.544 | 0.571 |
|           | Verification(1992-1998) | 0.6 | 0.56 | 0.5 |



**Table 2: Characteristics of droughts in discharge in the Eskandari catchment in the period with human influence (TLM=80[th]% natural period)**

| Drought type | No. drought | Duration(day) | | Deficit(mm) | | Human effect (%) | |
|---|---|---|---|---|---|---|---|
| | | max | mean | max | mean | Negative | Positive |
| Climate-induced drought | 41 | 204 | 62 | 10.3 | 2.4 | - | - |
| Human-induced drought | 45 | 192 | 72.9 | 9.2 | 2.4 | - | - |
| Human-modified drought | 39 | 306 | 115.2 | 21.9 | 4.1 | 96.9 | 3.1 |





**Table 3: Characteristics of droughts in discharge in the Kiakola catchment in the period with human influence (TLM=80[th]% natural period)**

| Drought type | No. drought | Duration(day) | | Deficit(mm) | | Human effect (%) | |
|---|---|---|---|---|---|---|---|
| | | max | mean | max | mean | Negative | Positive |
| Climate-induced drought | 34 | 173 | 42.6 | 20.3 | 2 | - | - |
| Human-induced drought | 31 | 115 | 32.7 | 3.5 | 0.7 | - | - |
| Human-modified drought | 40 | 177 | 43.9 | 9.8 | 1.1 | 88.9 | 11.1 |



**Table 4: results of drought in groundwater in Eskandari (threshold=80th percentile)**

| Type of anomaly | No. drought | Duration(month) | | Maximum deviation (m) | | Human effect (%) | |
|---|---|---|---|---|---|---|---|
| | | max | mean | max | mean | Negative | Positive |
| Climate-induced drought | 6 | 23 | 9 | 1.5 | 0.64 | - | - |
| Human-induced drought | 6 | 100 | 23.4 | 14.6 | 7.2 | - | - |
| Human-modified drought | 6 | 196 | 37.3 | 15.6 | 3.8 | 100 | 0 |





**Table 5: Characteristics of human-modified discharge and groundwater droughts in the Eskandari and Kiakola catchments in the period with human influence (threshold=50th, 70th and 90th percentile, natural period)**

| Catchment | Variable | Threshold (percentile) | No. drought | Duration(day) | | Deficit (mm) for discharge /maximum deviation for groundwater (m) | | Human effect % | |
|---|---|---|---|---|---|---|---|---|---|
| | | | | Max | Mean | Max | Mean | Negative | Positive |
| Eskandari | Discharge | 50th | 57 | 1250 | 128.4 | 164.2 | 10.8 | 79.2 | 20.8 |
| | | 70th | 42 | 1041 | 126.3 | 76.8 | 5.7 | 90 | 10 |
| | | 90th | 33 | 267 | 116.7 | 13.2 | 3 | 99 | 1 |
| Eskandari | Groundwater | 50th | 4 | 234 | 71.3 | 17.4 | 6.6 | 82.1 | 17.9 |
| | | 70th | 6 | 218 | 42.3 | 16.2 | 3.8 | 97.6 | 2.4 |
| | | 90th | 4 | 195 | 52.3 | 15 | 5 | 100 | 0 |
| Kiakola | Discharge | 50th | 75 | 189 | 46.6 | 24.1 | 3.6 | 78.8 | 21.2 |
| | | 70th | 48 | 179 | 43.8 | 13.9 | 1.8 | 89.1 | 10.9 |
| | | 90th | 32 | 146 | 35.1 | 4.7 | 0.4 | 95.9 | 4.1 |


**Figure 1: The step-wise approach to identify positive and negative human-modified droughts.**



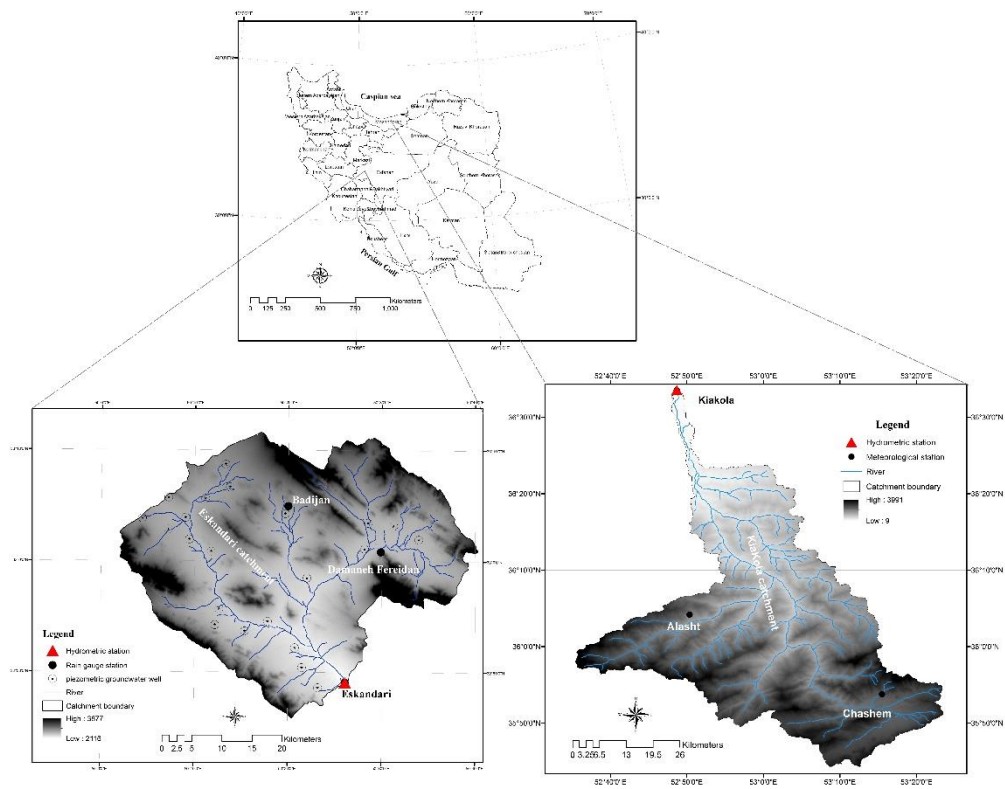

**Figure 2: Location of the Eskandari catchment (left) and Kiakola catchment (right) in Iran, including the topography, river network, meteorological station, gauging station, groundwater wells**

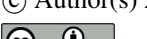



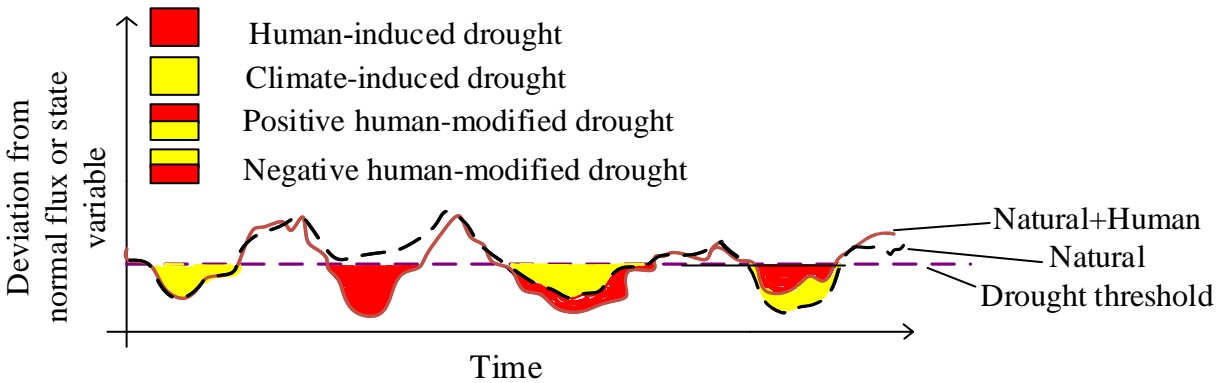

**Figure 3: Location of the catchments in Iran, including the topography, river network, meteorological station, gauging station, groundwater wells.**



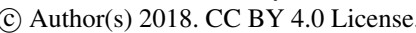



**Figure 4: Observed and simulated daily discharge in Eskandari (upper) and Kiakola (lower) for the natural period.**



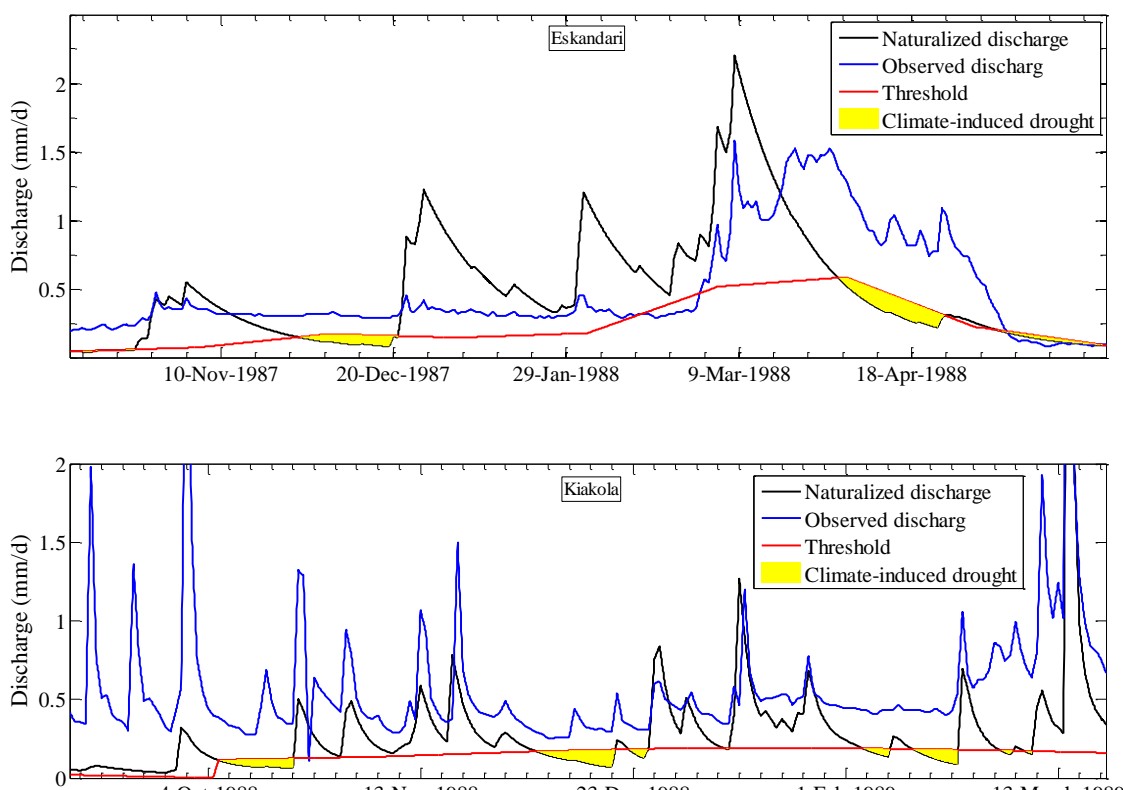

**Figure 5: Examples of climate-induced droughts in the Eskandari and Kiakola catchments**



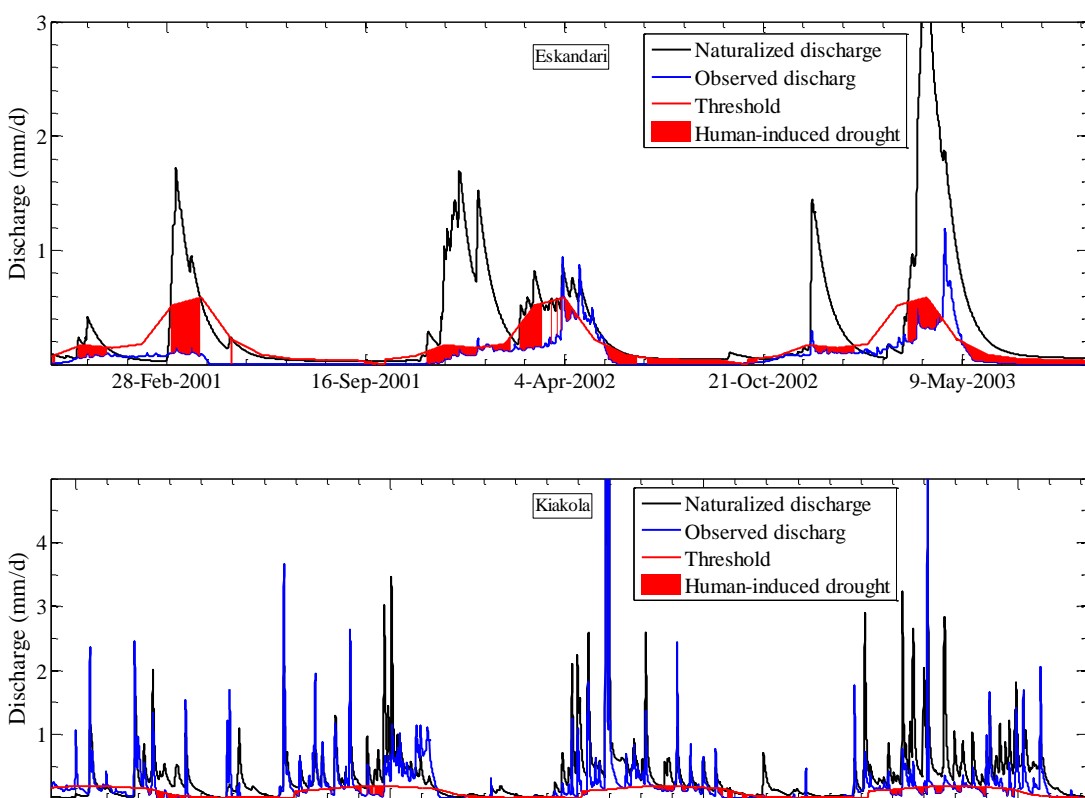

**Figure 6: Examples of human-induced droughts in the Eskandari and Kiakola catchments.**




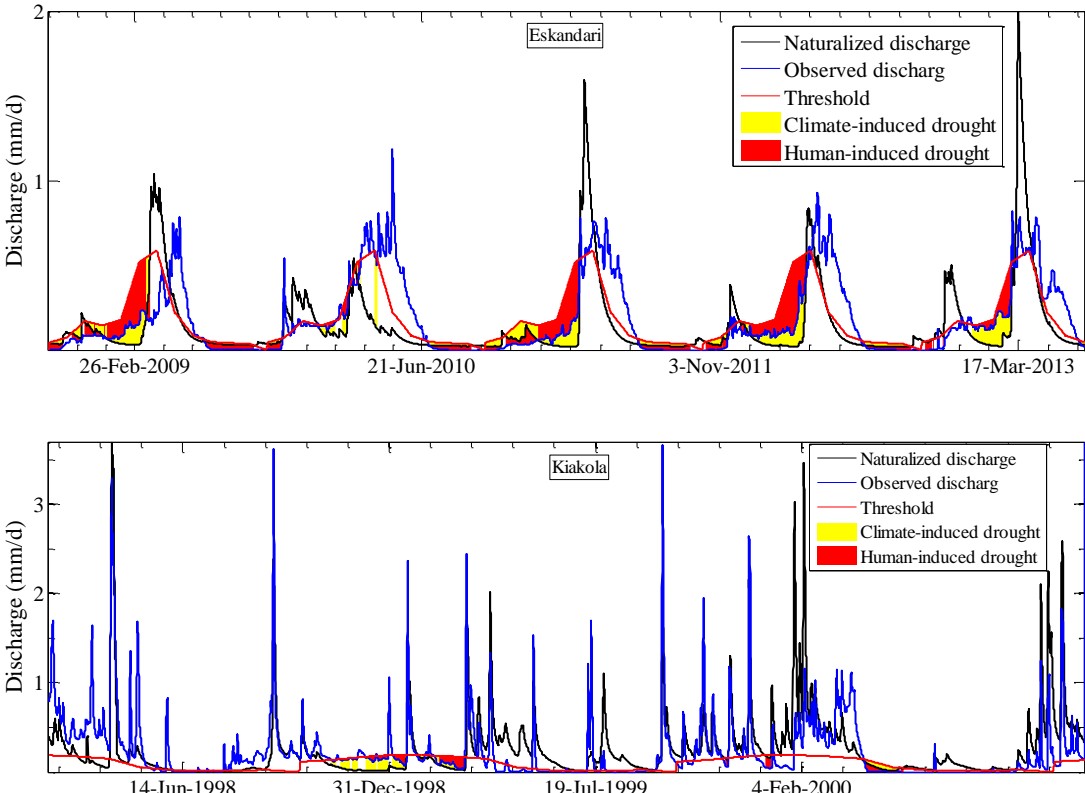

**Figure 7: Examples of human-modified droughts in the Eskandari and Kiakola catchments**




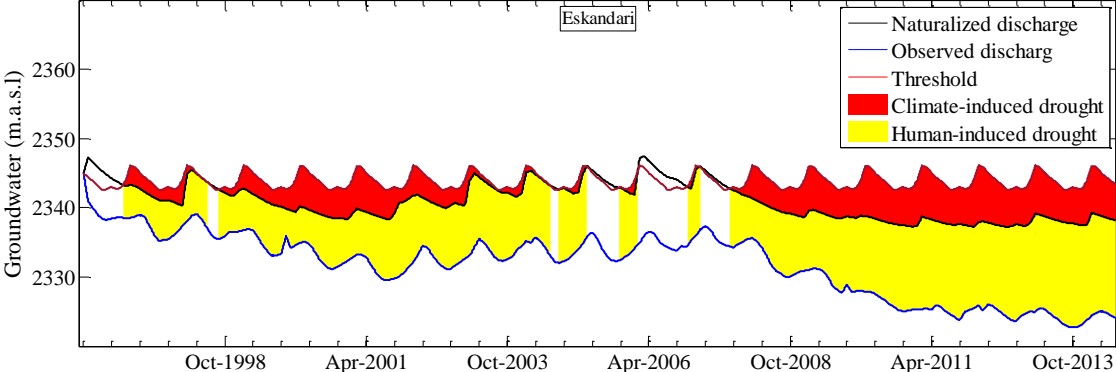

**Figure 8 Groundwater droughts in the Eskandari catchment**





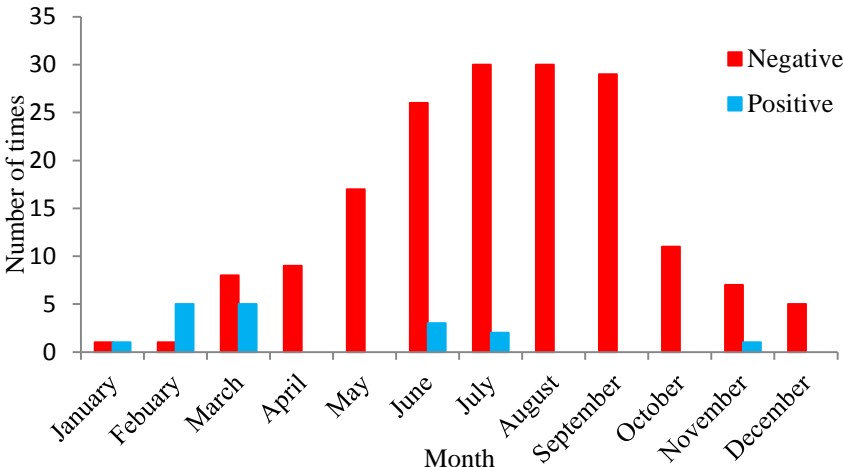

**Figure 9: Distribution of the negative and positive of human-modified droughts over the year in the Eskandari catchment (threshold=80$^{th}$ percentile)**





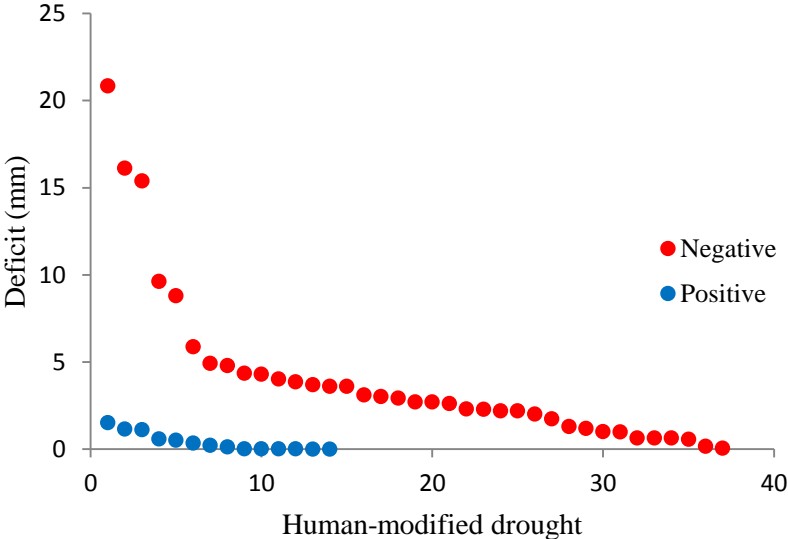

**Figure 10: Deficit volumes of the negative and positive of human-modified droughts in the Eskandari catchment (threshold=80th percentile)**



# Annex 1

**Table 1: Mann-Kendall test results for the Eskandari and Kiakola catchments**

| Catchment | Variable | Kendall's tau | p-value | Kendall test positive significance | Sen's slope estimate |
|---|---|---|---|---|---|
| Eskandari | Discharge | -0.176 | **<0.0001** | 1% | -6.878E-6 |
| | Rainfall | 0.033 | 0.043 | NS | 3.435E-5 |
| | Temperature | 0.022 | **< 0.0001** | 1% | 7.614E-5 |
| | Evapotranspiration | 0.014 | 0.017 | NS | 2.011E-5 |
| Kiakola | Discharge | -0.079 | **< 0.0001** | 1% | -1.201E-4 |
| | Rainfall | 0.073 | **< 0.0001** | 1% | 1.259E-4 |
| | Temperature | 0.053 | **< 0.0001** | 1% | 1.460E-4 |
| | Evapotranspiration | 0.016 | 0.014 | NS | 2.192E-5 |

Highlighted and bold values are indicating clear significance trends; NS: no significance.



## Annex 2:

**Table 1: Change point detection in discharge time series by using Pettitt's test for the Eskandari and Kiakola catchments**

| Catchment | p-value | alpha | Change point |
|-----------|---------|-------|--------------|
| Eskandari | < 0.0001 | 0.01 | May-1996 |
| Kiakola | < 0.0001 | 0.01 | May-1998 |

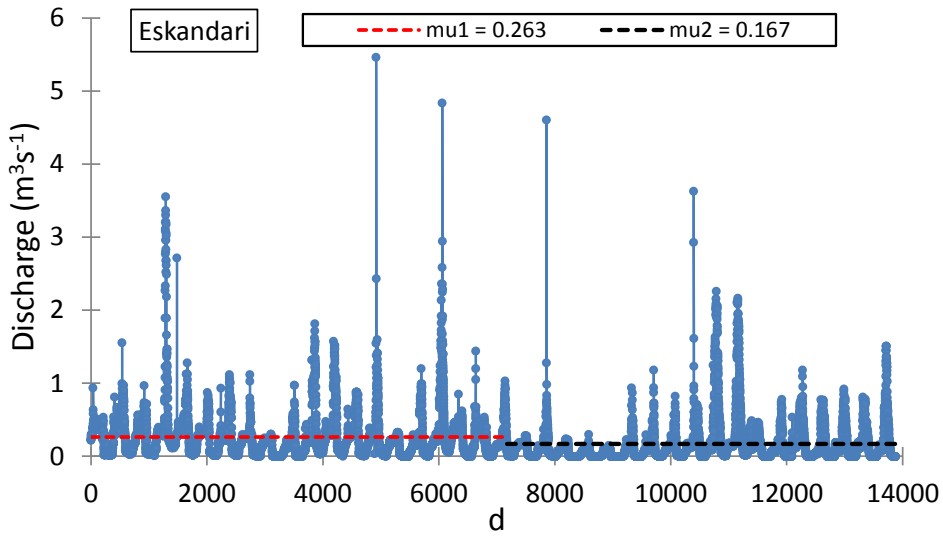

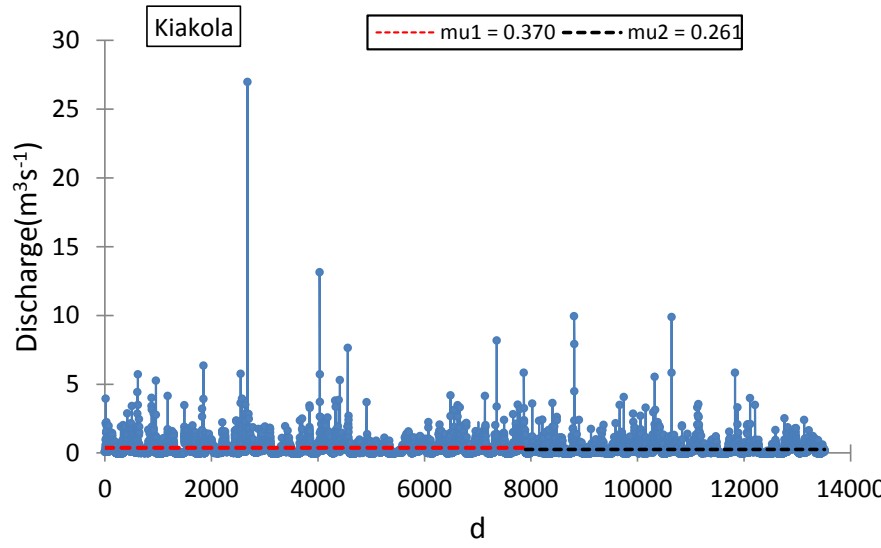

**Figure 1: Change point detection in discharge time series (top: Eskandari, down: Kiakola). Dashed dark blue and red lines are distribution of data before and after the change point**



# Annex 3:

**Table 1: HBV model parameters and the value ranges (Delay response routine)**

| Parameter | Explanation | Unit | Initial |
|-----------|-------------|------|---------|
| Snow routine | | | |
| TT | Threshold temperature | °C | (−2.5  2.5) |
| CFMAX | Degree-day factor | $mm°\text{C}^{-1}d^{-1}$ | (1  10) |
| SFCF | Snowfall correction factor | - | (0.2  0.6) |
| CFR | Refreezing coefficient | - | (0  0.05) |
| CWH | Water holding capacity | - | (0  0.5) |
| Soil moisture routine | | | |
| FC | Maximum SM (storage in the soil) | $mm$ | (70  300) |
| LP | Threshold for reduction of evaporation (SM/FC) | - | (0.3  1) |
| BETA | Shape coefficient | - | (1  2) |
| Response routine | | | |
| Alpha | Nonlinearity parameter | - | (0  1) |
| K1 | Recession coefficient | $d^{-1}$ | (0.02  0.2) |
| K2 | Recession coefficient | $d^{-1}$ | (0.0005  0.1) |
| Routing routine | | | |
| MAXBAS | Routing, length of weighting function | $d$ | (1  10) |
| Other | | | |
| PART | the portion of the recharge which is added to the groundwater box | - | (0.5  1) |
| DELAY | period of delay | $d$ | (1  50) |





**Table 2: comparison of 50th and 80th percentile of the duration curves of flow discharge and groundwater (Eskandari catchment)**

| | | | Jan | Feb | Mar | Apr | May | Jun | Jul | Aug | Sep | Oct | Nov | Dec | Annual |
|---|---|---|---|---|---|---|---|---|---|---|---|---|---|---|---|
| Eskandari | $Q_{sim}$ | 50% | 0.25 | 0.24 | 0.41 | 0.62 | 0.13 | 0.05 | 0.03 | 0.02 | 0.04 | 0.14 | 0.22 | 0.27 | 0.18 |
| | | 80% | 0.17 | 0.2 | 0.33 | 0.25 | 0.02 | 0.01 | 0.02 | 0.01 | 0.02 | 0.08 | 0.13 | 0.2 | 0.023 |
| | $Q_{obs}$ | 50% | 0.25 | 0.25 | 0.38 | 0.65 | 0.13 | 0.04 | 0.02 | 0.02 | 0.03 | 0.14 | 0.22 | 0.26 | 0.18 |
| | | 80% | 0.16 | 0.21 | 0.28 | 0.29 | 0.04 | 0.01 | 0.01 | 0.01 | 0.004 | 0.09 | 0.14 | 0.19 | 0.02 |
| | $GW_{sim}$ | 50% | 21.6 | 20.6 | 20 | 20 | 20.3 | 20.7 | 21 | 21.3 | 21.7 | 21.8 | 21.6 | 21.5 | 20.99 |
| | | 80% | 18.7 | 16.5 | 14.8 | 15.6 | 16.2 | 16.8 | 17 | 17.9 | 18.3 | 17.7 | 17.7 | 18.1 | 18.08 |
| | $GW_{obs}$ | 50% | 21.8 | 21.2 | 20.6 | 19.9 | 20.2 | 21.3 | 22.1 | 22.8 | 22.9 | 22.6 | 22.5 | 22.1 | 21.7 |
| | | 80% | 20 | 19.7 | 19.1 | 18.5 | 18.3 | 19.2 | 20 | 20.7 | 21.2 | 21.1 | 21.1 | 20.4 | 19.8 |
| Kiakola | $Q_{sim}$ | 50% | 0.39 | 0.59 | 0.67 | 0.17 | 0.05 | 0.03 | 0.03 | 0.23 | 0.11 | 0.18 | 0.24 | 0.29 | 0.25 |
| | | 80% | 0.24 | 0.45 | 0.53 | 0.05 | 0.004 | 0.01 | 0.01 | 0.02 | 0.04 | 0.09 | 0.11 | 0.15 | 0.04 |
| | $Q_{obs}$ | 50% | 0.35 | 0.53 | 0.63 | 0.13 | 0.03 | 0.1 | 0.02 | 0.1 | 0.14 | 0.19 | 0.24 | 0.32 | 0.22 |
| | | 80% | 0.21 | 0.35 | 0.42 | 0.04 | 0.003 | 0.03 | 0.01 | 0.01 | 0.07 | 0.12 | 0.14 | 0.19 | 0.03 |



# Annex 4:

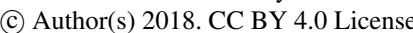


5        **Figure 1: examples of climate-induced drought, human-induced drought and human modified drought for Eskandari**





**Figure 2: examples of climate-induced drought, human-induced drought and human modified drought for Kiakola**

