# Peer review of "Positive and negative human-modified droughts: a quantitative approach illustrated with two Iranian catchments"

_Hydrology and Earth System Sciences, 2018_

## Referee Comment (RC1) · Anonymous Referee #1 · 29 Jun 2018

The paper investigates human-modified droughts; an interesting topic in the scope of the journal and at least theoretically well supported by the background literature in the introduction. The analysis is based on a framework recently proposed by van Loon et al. (2016a,b) to distinguish between climate- and human-induced and human-modified drought events and to disentangle the causes for drought, namely climate variability and anthropogenic forcing due to water abstractions from rivers and groundwater. The method differentiates between periods of natural and disturbed streamflow and utilized then a N-A-model (calibrated during the natural period) to calculate natural streamflow during the proposed period of disturbed streamflow. In principle the paper proposes that different drivers of streamflow droughts can be counted against each other to gain

a cumulative drought signal. The authors concluded that for the two presented catchments human activities have caused mostly negative human-modified droughts.

Although the research topic is important I doubt that structure of the methodology, the presentation of the results and their interpretation lead to clear conclusions and advanced implications. I missed a clear and structured development and description of the method. I think the editor has to decided whether major revisions and structural and graphical improvements are needed or a new submission of the paper is the better way. I encourage the authors to revise the method description, the analysis and the graphs to gain a more comprehensive paper.

+ STUDY CONCEPT As the study concept (components of droughts) is rather new I suggest to explain the different types in more detail. Why were exactly these two catchments chosen for the analysis and what are the "major concern" (p2l32) in detail regarding the human-modified droughts here? Section 2.2.1 and 2.2.2 could be shorten (well-known statistical tests), but more justification is needed to proof that e.g. Pettitt's Test is really appropriate to distinguish periods of natural and disturbed streamflow (e.g., change points can also emerge due to climate change). Also the explanation of the anomaly analysis isn't straightforward for me; what is surplus value of all these equations and subscripts (eq. 8-11) if the concept could be explained with a good overview figure?

+ HBV MODELING For me it is not possible to evaluate whether the model is appropriate to answer the research questions or not. For example Fig.6. shows huge discrepancies in observed and simulated streamflow (btw: Why is the black line here needed if the observed discharge is only compared to the threshold?). The authors stated that the model performance is appropriate, but in my eyes the model performance is rather poor suggesting that the model setup / preparation / calibration has some issues. Is NSE or logNSE used (it's not clear from the text, p7l6)? However, is NSE really the best OF or would be a measure with more emphasis on volume errors a better choice? How valuable are the results (Tab.2+3) showing mean drought deficits

of 0.7 mm to 4.1 mm if they can easily attributed to model simulation uncertainty? The different thresholds in Tab.5 somehow shows that higher thresholds (50th) are needed to gain significant deficit volumes in respect to drought management. If so, I ask myself if a 50th threshold is really valuable for drought assessment?

+ DATA I understand that these kind of studies in data-scare and semi-arid regions are very important for local water management. However, in this case a relatively novel method is used to investigate different drought types. Would it be better to have a larger (and perhaps better) data set to eliminate all the catchment-specific issues of the two proposed catchments? At least the method could be conducted on a larger data set including the two Iran catchments. However, if it is really important to understand climate- and human-influence in these catchment, then a clear justification for the change points in the time series have to be made (e.g., due to better visualizations of the change).

+ GRAPHS The quality and in-depth information of the single graph is often below average. Fig.2 is hardly readable due to graph quality issues. For me it is not possible with Fig 4. to see how good/or bad the model is performing, the lines are too tick, the axis labels are not appropriate (here more in-depth analysis is needed, e.g. FDCs), Fig.5-7 show examples but no systematic visualization of different drought types, Fig.1 (Annex) try to highlight the change point in the time series, but the reader cannot find any justification for this statement (improvement: remove the points, check if y-log-scale would be helpful, compare annual streamflow series from the period before/after the proposed change point). Overall some synthesis graphs are really missing to support the points the authors want to make.

---

## Referee Comment (RC2) · Anonymous Referee #2 · 11 Jul 2018

This paper proposes a new methodology to identify positive and negative human modified droughts and tests the methodology on two Iranian catchments. There certainly should be more studies investigating human-modified droughts and it is refreshing to see a case study application with data from catchments in Iran.

However, there are a number of key issues with the study that affect the robustness of the results and conclusions. While I agree with the authors that the methodology could enable quantification of positive and negative human modified drought in the Anthropocene - currently (1) the methods are not very clear which makes it difficult to understand exactly what was done in the study and (2) the methodology only works and

attributes droughts correctly if the modelling simulations are robust. The model performance and large differences during the naturalized period do not provide confidence in the validity of the key results of the paper and instead I believe that the human-modified droughts identified are more due to hydrologic model uncertainty rather than human activities. The presentation of the figures also needs to be significantly improved. I encourage the authors to re-address the hydrological modeling they have done, to significantly revise the methods section and to improve the quality of the figures to support their key results.

Main Comments

Subdividing the time series into natural and disturbed

Better justification is needed to prove that these statistics are identifying change points. One way could be to gain qualitative data about human activity that is occurring in the catchment (i.e. was a dam built in the change-point year, or a significant new water abstraction implemented?). Figure 1 also needs improving- currently it is really difficult to see any evidence of a distinct change point. As low flows are of interest then discharge could be plotted on a log scale, you should remove the blue dots and I would also add precipitation data. I would also consider plotting a much shorter timeseries (perhaps years that have similar climatic characteristics for both the natural and disturbed period).

HBV Hydrological Modelling Set Up

a) Objective Function – It isn't clear from the paper whether you used logNSE or NSE. There also needs to be better justification for this use of objective function – there are lots of different metrics that evaluate low flows and you need to better justify your choice here and discuss how it might impact the results.

b) Calibration – The calibration section needs to be much better explained. On Page7, L6 it states that 'The calibration procedure was done more than fifty times' and then

on L9 it states 'each calibration was repeated 100'. It isn't particularly clear what the calibration procedure is (simply optimizing logNSE but then R2 is introduced?), how the parameters ranges were chosen for each of the 14 parameters, how many samples were implemented (50 or 100?) etc. On L9 it also states that this 'resulted in 100 different parameterizations' but it isn't clear how these 100 different parameterizations were used in the rest of the study – I assume you just took the best one to use in the rest of the study but given the uncertainty in the hydrological model, your results would be much more robust if you calculated natural-human drought characteristics for all 100 parameterizations.

c) Calibration Results – It is very difficult to see from Figure 4 the comparison between the observed and simulated discharge. It would be useful to plot a comparison of the flow duration curves (on log scale)

HBV Model Results

The attribution of positive and negative human-modified droughts rests entirely on the performance of the model and it's ability to represent the naturalised flows. From Figure 5, the observed flows are plotted against the modelled naturalised flows for the period 1987-1988 for Eskandari and 1988 – 1989 for Kiakola. As I understand it from Section 2.2.2 both these periods lie in the 'natural' period (undisturbed period) for each catchment and so you would expect the naturalised model flow to (as much as possible) represent the observed flow and so climate-induced droughts to be identified at the same point in the time series. In this case, there are quite large discrepancies between the observed and modelled naturalised flows and climate-induced droughts are identified at completely different points. This is because the threshold is derived from the observed flows (which are very different to the naturalised flows produced by the model) and so casts doubts on the modelling results as they should be similar for the naturalised period. This has knock-on impacts for the attribution of positive and negative human modified droughts, which again I think are due to model simulation uncertainty rather than human influences. Consequently, it is difficult to have confidence

in the results and key conclusions of the paper.

---

## Author Comment (AC2) · 25 Sep 2018

Interactive comment on "Positive and negative human-modified droughts: a quantitative approach illustrated with two Iranian catchments"

by Elham Kakaei et al.

Anonymous Referee #1

The paper investigates human-modified droughts; an interesting topic in the scope of the journal and at least theoretically well supported by the background literature in the introduction. The analysis is based on a framework recently proposed by van Loon et

al. (2016a,b) to distinguish between climate- and human-induced and human-modified drought events and to disentangle the causes for drought, namely climate variability and anthropogenic forcing due to water abstractions from rivers and groundwater. The method differentiates between periods of natural and disturbed streamflow and utilized then a N-A-model (calibrated during the natural period) to calculate natural streamflow during the proposed period of disturbed streamflow. In principle the paper proposes that different drivers of streamflow droughts can be counted against each other to gain a cumulative drought signal. The authors concluded that for the two presented catchments human activities have caused mostly negative human-modified droughts. Although the research topic is important I doubt that structure of the methodology, the presentation of the results and their interpretation lead to clear conclusions and advanced implications. I missed a clear and structured development and description of the method. I think the editor has to decide whether major revisions and structural and graphical improvements are needed or a new submission of the paper is the better way. I encourage the authors to revise the method description, the analysis and the graphs to gain a more comprehensive paper.

- The authors appreciate the reviewer's constructive comments. The comments have been incorporated in the revised manuscript. Responses to the specific comments can be found below. We trust that the revisions we propose support better the conclusions and the implications. We think that the stepwise approach from Step 1 to Step 5 (Sections 2.2-2.5), which is presented in Figure 1, provides a well-elaborated description of the methodology.

Main comments

+ STUDY CONCEPT As the study concept (components of droughts) is rather new I suggest to explain the different types in more detail. Why were exactly these two catchments chosen for the analysis and what are the "major concern" (p2l32) in detail regarding the human-modified droughts here? Section 2.2.1 and 2.2.2 could be shorten (well-known statistical tests), but more justification is needed to proof that e.g.

Pettitt's Test is really appropriate to distinguish periods of natural and disturbed stream-flow (e.g., change points can also emerge due to climate change). Also the explanation of the anomaly analysis isn't straightforward for me; what is surplus value of all these equations and subscripts (eq. 8-11) if the concept could be explained with a good overview figure?

Response: - The different types of drought, that is, climate-induced drought, human-induced drought and human modified drought are explained in more detail in Section 2.5 (P8, l12-31 and P9, l1-19).

- The Eskandari and Kiakola catchments in Iran have been chosen, because we would like to have catchment from a dry climate that face climate-induced drought and usually have to cope with human interventions that cause the drought characteristics to change, either in a positive or in a negative way, i.e. positive human-modified drought or negative human-modified drought, respectively. The drawback is that catchments from dry environments often are data sparse. We anticipated that this is acceptable because the focus of the paper is to propose a methodology to break down the human-modified droughts into positive and negative ones. We have added a line in the revised manuscript to explain this (P2, l30-33, P3, 127-129).

- The main concerns about human intervention in both Iranian catchments are land use change, incl. crop pattern change, large groundwater abstraction and population growth. We have explained these human interventions in Annex 1 to this reply. We suggest not to include all these details in the main text of the paper, because the focus of the paper is on the methodology, as said above. Alternatively, we can add the text as an Annex to the paper.

- We agree with the reviewer that the description of the well-known statistical tests can be shortened. Hence, Sections 2.2.1 and 2.2.2 are made briefer, and furthermore a more-elaborated justification is presented to proof Pettit's test ability to distinguish between natural and disturbed periods. The results of Pettit's test for rainfall time series in

both catchment are added to Annex 2 of the paper (P37, Table 1) and (P38, Figure 2). Furthermore FDCs of monthly observed discharge records were plotted for the natural and disturbed periods (P39, Figure 3). As explained in Section 3.1 (P10 l22-l27 and P11 l1-l7), in the Eskandari catchment, the change point in discharge and rainfall time series approximately occurred in May 1996 and October 2001, respectively. In addition, according to the Mann-Kendall test (Section 3.1, P10 l5-l22), the discharge showed a clear significant downward trend, while, rainfall showed a weak none-significant upward trend, which indicate the influence of both natural (weather variables) and human drivers on discharge. Based on the Pettit's test the period 1976-1996 could be classified as the natural period (undisturbed period) and the period 1996-2014 as disturbed. In the Kiakola catchment, the change point in discharge and rainfall time series roughly occurred in May 1998 and September 2004, respectively. Considering the result of Pettit's test on discharge and rainfall and the Mann-Kendall test results in Section 3.1 (P 10 l5-l22) (significant downward trend in discharge, significant upward trend in rainfall, and a none-significant upward trend in evapotranspiration), the periods 1976-1998 and 1998-2013 could be considered as natural and disturbed, respectively. For both catchments (Eskandari and Kiakola), the change point in rainfall time series occurred 5 years (2001) and 6 years later (2006) than changes in discharge (1996 and 1998, respectively). So, the change point in discharge has mainly to be contributed to human intervention in both catchments. We would like to stress here that the application of statistical tests is important in the overall methodology to distinguish between the natural and disturbed periods (Step 1). The actual results for the two Iranian catchment are less important, as mentioned above. These only illustrate what need to be achieved in Step 1.

- The equations of climate-induced drought, human-induced drought and human-modified drought formalize the definitions of each type of human-modified drought and complement the conceptual diagram Figure 3. We believe that these equations have added value and leave no room for different interpretation. The equations expand on those presented by Tallaksen et al. (2009). We added some text to clarify this (P8

l25-l31 and P9 l1- l10).

+ HBV MODELING For me it is not possible to evaluate whether the model is appropriate to answer the research questions or not. For example Fig.6 shows huge discrepancies in observed and simulated streamflow (btw: Why is the black line here needed if the observed discharge is only compared to the threshold?). The authors stated that the model performance is appropriate, but in my eyes the model performance is rather poor suggesting that the model setup / preparation / calibration has some issues. Is NSE or logNSE used (it's not clear from the text, p7l6)? However, is NSE really the best OF or would be a measure with more emphasis on volume errors a better choice? How valuable are the results (Tab.2+3) showing mean drought deficits of 0.7 mm to 4.1 mm if they can easily attributed to model simulation uncertainty? The different thresholds in Tab.5 somehow shows that higher thresholds (50th) are needed to gain significant deficit volumes in respect to drought management. If so, I ask myself if a 50th threshold is really valuable for drought assessment.

Response:

- The natural time series is shown in Figure 6 (P31) to reflect natural conditions without human drivers of drought. The natural time series are an essential component of identifying human-modified droughts (Van Loon et al., 2016ab; Van Loon and Van Lanen, 2012).

- The model performance is evaluated based on the lnNSE, which is an appropriate objective function for low flow simulation (P7 l9-l21). It has been frequently used. Each of the statistical criteria has specific pros and cons, which have to be taken into account during model calibration and evaluation. The Nash-Sutcliffe efficiency (NSE) and the coefficient of determination are very sensitive to peak flows, at the expense of being less sensitive to low flow conditions. They are based on the squared differences between observed and simulated values (Pushpalatha et. al., 2012; Krause et. al., 2005). Additionally, the coefficient of determination alone should not be used for model evaluation, because it can have still high values for very poor model results, because it is based on the correlation only. In order to reduce the problem of the squared differences and the resulting sensitivity to high extreme values, the Nash-Sutcliff efficiency is often calculated with logarithmic values of observed and simulated values (lnNSE), in particular if drought is considered. Through the logarithmic transformation of the discharge values the peaks are flattened and low flows are kept more or less at the same level. As a result the influence of the low flow values is increased in comparison to the peaks values resulting in an increase in sensitivity of lnNSE to systematic model over- or underestimation. So, the lnNSE reacts less on peak flows and stronger on low flows than the NSE (Krause et. al., 2005). So, the logarithmic Nash-Sutcliffe efficiency according to the algorithm of the observed and simulated flow discharge (lnNSE), as the best objective function for low-flow modelling, was utilized for the assessment of discrepancies between observed and simulated discharge and the performance of the HBV in both catchment

- The NSE$\geq$0.5 has been defined as an acceptable value for model performance (Christiansen, 2012; Moriasi et. al., 2007). Moreover, we would like to stress again that the focus of the paper is on the methodology rather than on the actual outcome for the Eskandari and the Kiakola catchments. Hence, model performance is not of uttermost importance (P7, l22-l24), as long as it passes the minimum criterion.

- Mean drought deficits are small, because minor droughts with small deficits are included as well, although events <15 days were excluded from the analysis. Clearly, model uncertainty will affect the numbers, but outcome is still valuable because other model structures will result in similar relative small numbers. As said before, the focus of the paper is on the methodology rather than on the exact numbers for the two Iranian catchments. These are only in the study to demonstrate the methodology.

- Selection of the threshold level is quiet challenging. For drought management, not only the yearly recurring (summer or winter) low-flow period is important, but any deviation from the normal seasonal pattern (Van Loon and Van Lanen, 2012). Most of

the time in arid and semi-arid regions the discharge is zero or close to zero and the time series of the data sets (discharge and rainfall) contain a significant number of zeros values. In the Iranian study regions, there is a strong seasonality in rainfall and discharge regimes and it is not uncommon to find extended dry periods up to several months where no rainfall is observed, i.e. zero values (Favier et al., 2009; Verbist et al., 2010; Meza, 2013).The threshold of 50th percentile could have been used to identify drought events to avoid a threshold of zero (Rangecroft et al., 2016; Giannikopoulou et al., 2014; Van Huijgevoort et al., 2012). We tested the effect of using different threshold values on drought characteristics (Section 4), and eventually on the percentage of negative and positive human-modified droughts (Step 5) by using the 50th, 70th and 90th percentiles. As hypothesized, choosing different percentiles as threshold will change the magnitude of the drought characteristics. A lower threshold (90th percentile) leads to fewer human-modified events with shorter durations and lower deficits, whereas, higher thresholds (50th and 70th percentiles) results in the opposite. Eventually, the magnitude of the threshold has to be defined by the drought manager or stakeholders. For perennial streams thresholds of 70-90th are commonly used. In semi-arid and arid climates (e.g. Iran), higher thresholds, e.g. 50th, could also be used and provide relevant information (Fleig et al., 2006). In the paper we have selected the 80th percentile, because it has been used in most drought studies, but we also included higher and lower thresholds. This has not been done to discuss drought management (not the aim of the paper), but because we would like to show that the selection of the threshold does not affect the pattern of negative and positive human-modified drought (which is the purpose of the paper).

+ DATA I understand that these kind of studies in data-scare and semi-arid regions are very important for local water management. However, in this case a relatively novel method is used to investigate different drought types. Would it be better to have a larger (and perhaps better) data set to eliminate all the catchment-specific issues of the two proposed catchments? At least the method could be conducted on a larger data set including the two Iran catchments. However, if it is really important to understand climate- and human-influence in these catchment, then a clear justification for the change points in the time series have to be made (e.g., due to better visualizations of the change).

Response: - The paper proposes a step-wise methodology to expand on the distinction between different hydrological drought types in the Anthropocene, in particular making a distinction between positive human-modified and negative human-modified droughts. The proposed methodology, as a multi-directional and multi-driver drought framework, enables a further elaboration of drought in the Anthropocene. So, it makes it possible for water resources managers to decide on how to combat natural and human-made droughts. Long time series of hydrometeorological data for both natural and disturbed period are the basic elements of the proposed methodology, irrespective of the region. We decided to use data from a dry environment, because climate-induced droughts are frequent there and human interventions are common, likely causing human-modified drought. We realized that many of these time series have quality issues (e.g. short time series, gaps). In addition, this kind of information is not available because of water conflicts and security reasons in many arid and semi-arid regions. However, we think that the data of the two Iranian catchments are sufficient to illustrate the potential of the proposed methodology. In a follow-up study a larger set of catchments can be analyzed to explore differences between different climate settings and human influences. The paper certainly has not the aim to understand in depth the climate and human influence on drought types in the two Iranian catchment.

+ GRAPHS The quality and in-depth information of the single graph is often below average. Fig.2 is hardly readable due to graph quality issues. For me it is not possible with Fig 4. to see how good/or bad the model is performing, the lines are too tick, the axis labels are not appropriate (here more in-depth analysis is needed, e.g. FDCs), Fig.5-7 show examples but no systematic visualization of different drought types, Fig.1 (Annex) try to highlight the change point in the time series, but the reader cannot find any justification for this statement (improvement: remove the points, check if y-log-scale

would be helpful, compare annual streamflow series from the period before/after the proposed change point). Overall some synthesis graphs are really missing to support the points the authors want to make.

Response:

- We revised a number of graphs. Figure 2 has been revised and hence the quality improved (P27). Figure 4 is replaced by the FDCs of observed and simulated discharge for the natural period (P29). Figures 5-7 are revised (P30-32). Figure 1 (Annex 2) has been deleted and the change point in the discharge and rainfall time series are put on a y-logarithmic scale (P37 Figure 1 and P38 Figure 2). In addition, the FDC of monthly discharge has been added for the natural and disturbed periods before and after the change point (P39 Figure 3).

- We think that the synthetic graph (Figure 3) together with the equations adequately present the generic points we would like to make.

References Christiansen, D.E.: Simulation of daily streamflows at gaged and ungaged locations within the Cedar River Basin, Iowa, using a Precipitation-Runoff Modeling System model: U.S. Geological Survey Scientific Investigations Report 2012–5213, 20 p, 2012. Favier, V., Falvey, M., Rabatel, A., Praderio, E., López, D. Interpreting discrepancies between discharge and precipitation in high-altitude area of Chile's Norte Chico region (26-32°S). Water Resour Res, 45, W02424, 2009. Fleig, A. K., L. M. Tallaksen, H. Hisdal, and Demuth, S.: A global evaluation of streamflow drought characteristics, Hydrol. Earth Syst. Sci., 10, 535–552, doi:10.5194/hess-10–535-2006, 2006 Giannikopoulou, A. S., Kampragkou, E., Gad, F. K., Kartalidis, A., Assimacopoulos, D. Drought characterisation in Cyclades complex, Greece. European Water, 47: 31 – 43, 2014. Gohari, A., Madani, K., Mirchi, A., and Bavani, A.M.: System-Dynamics approach to evaluate climate change adaptation strategies for Iran's Zayandeh-Rud Water System. In: Proceedings of the World Environmental and Water Resources Congress 2014, 1598–1607, 2014. Gohari, A., Mirchi, A., and Madani, K.: System

dynamics evaluation of climate change adaption strategies for water resources management in central Iran, Water Resour Manage, 31, 1413–1434, doi: 10.1007/s11269-017-1575-z, 2017. Hesheminya, S.M., and Dehghannya, J.: Investigation of basic factors influencing rice losses in Iran, Int J Agri Crop Sci, 5, 2190-2192, 2013. Heshemy Shahdany, S.M., and Firoozfar, A.R.: Providing water level control in main canals under significant inflow fluctuations at drought periods within canal automation, Water Resources Management, 31, 3343-3354, https://doi.org/10.1007/s11269-017-1671-0, 2017. Kavian, A., Mohammadi, M., Gholami, L., and Rodrigo-Comino, J.: Assessment of the Spatiotemporal Effects of Land Use Changes on Runoff and Nitrate Loads in the Talar River, 10, 445, doi:10.3390/w10040445, 2018. Krause, P., Boyle, D.P., and Bäse. F.: Comparison of different efficiency criteria for hydrological model assessment, Advances in Geosciences, 5, 89–97, https://doi.org/10.5194/adgeo-5-89-2005, 2005. Madani, K.: Water management in Iran: what is causing the looming crisis?, Journal of Environmental Studies and Sciences, 4, 315–328, https://doi.org/10.1007/s13412-014-0182-z, 2014. Madani Larijani, K.: Iran's water crisis: inducers, challenges and counter-measures. Proceedings of the ERSA 45th Congress of the European Regional Science Association, Vrije University, Amsterdam, The Netherlands, August 2005. Madani, K., and Marino, M.A.: System dynamic analysis for managing Iran's Zayandeh-Rud river basin, Water resources management, 23, 2163-2178, doi: 10.1007/s11269-008-9376-z, 2009. Meza, F. J. Recent trends and ENSO influence on droughts in Northern Chile: An application of the Standardized Precipitation Evapotranspiration Index. Weather and Climate Extremes, 1: 51 – 58, 2013. Mirzaei, M., Solgi, E., and Salmanmahiny, A.: Assessment of impacts of land use changes on surface water using L-THIA model (case study: Zayanderud river basin), Environ Monit Assess, 188-690, doi: 10.1007/s10661-016-5705-5, 2016. Moriasi, D.N., Arnold, J.G., Van Liew, M.W., Bingner, R.L., Harmel, R.D., and Veith, T.L.: Model Evaluation Guidelines for Systematic Quantification of Accuracy in Watershed Simulations, Transactions of the ASABE, 50, 885–900, 2007. Pushpalatha, R., Perrin, C., Nicolas, L.M., and Andreassian, V.: A review of efficiency criteria suitable for evaluating low-flow simulations, Journal of

Hydrology, 420–421, 171–182, doi:10.1016/j.jhydrol.2011.11.055, 2012. Rangecroft, S., Van Loon, A. F., Maureira, H., Verbist, K., and Hannah, D. M.: Multi-method assessment of reservoir effects on hydrological droughts in an arid region, Earth Syst. Dynam. Discuss., https://doi.org/10.5194/esd-2016-57, 2016. Tallaksen, L.M., Hisdal, H. and Van Lanen, H.A.J.: Space–time modelling of catchment scale drought characteristics, Journal of Hydrology, 375, 363-372, doi: 10.1016/j.jhydrol.2009.06.032, 2009. Sarhadi, A., and Soltani, S.: Determination of waterrequirement of the Gavkhuni wetland, Iran: a hydrological approach, Journal of Arid Environments, 98, 27-40, http://dx.doi.org/10.1016/j.jaridenv.2013.07.010, 2013. Shafiee, A.H., and Safamehr, M.: Study of water resources system of Zayanderud dam through area increment and area reduction methods, Isfahan province, Iran, Procedia Earth and Planetary Science, 4, 29 – 38, doi: 10.1016/j.proeps.2011.11.004, 2011 Van Huijgevoort, M. H., Hazenberg, P., van Lanen, H. A. J. & Uijlenhoet, R. A generic method for hydrological drought identification across different climate regions. Hydrol Earth Syst Sc, 16: 2437 – 2451, 2012. Van Loon A., and Van Lanen H.: A process-based typology of hydrological drought, Hydrol. Earth Syst. Sci., 16 (7): 1915-1946, doi: 10.5194/hess-16-1915-2012, 2012. Van Loon, A. F., Gleeson, T., Clark, J., Van Dijk, A. I. J. M., Stahl, K., Hannaford, J., Di Baldassarre, G., Teuling, A. J., Tallaksen, L.M., Uijlenhoet, R., Hannah, D.M., Sheffield, J., Svoboda, M., Verbeiren, B., Wagener, T., Rangecroft, S., Wanders, N., and Van Lanen, H.A.J.: Drought in the Anthropocene, Nature Geosci.,9, 89–91, doi: 10.1038/ngeo2646 , 2016a. Van Loon, A.F, Stahl,K., Di Baldassarre,G., Clark, J., Rangecroft, S., Wanders, N., Gleeson, T., Van Dijk, A.I.J.M., Tallaksen, M., Hannaford, Jamie., Uijlenhoet, R., Teuling, A.J., Hannah, D.M., Sheffield, J., Svoboda, M., Verbeiren, B., Wagener, T., and Van Lanen, H.A.J.: Drought in a human-modified world: reframing drought definitions, understanding, and analysis approaches, Hydrol. Earth Syst. Sci., 20, 3631–3650, doi: 10.5194/hess-20-3631-2016, 2016b. Verbist, K., Robertson, A., Cornelis, W.M., Gabriels, D. Seasonal predictability of daily rainfall characteristics in central-northern Chile for dry-land management. J Appl Meteorol Clim, 49(9): 1938-1955, 2010. Zare, M., Nazari Samani, A.A., Mohammady, M., Salmani, H.,

and Bazrafshan, J.: Investigating effect of land use change scenarios on soil erosion using CULE-s and RUSLE models, International Journal of Environmental Science and Technology, 14, 1905–1918, https://doi.org/10.1007/s13762-017-1288-0, 2017.

Annex 1: Human interventions in the Eskandari and Kiakola catchments

During the past decades water resources management in the Zayandehrood river basin, which includes the Eskandari as an important tributary, has been a critical issue, because of the growing human population, industry, high agricultural water use and frequent droughts, land use change, incl. decrease of pasture land and forests (Mirzaei et. al., 2016; Gohari et. al. 2017). These have caused unprecedented water shortages in the basin. Meteorological and hydrological extreme events are common in this basin (Madani and Mariño, 2009). The Zayandehrood River, as the backbone of human development in central Iran, dries up seasonally, which imposes extensive pressure on the urban population, agriculture and industries (Madani Larijani 2005). Droughts in past decades has decreased the capacity of Zayandehrood dam's lake to less than 150 MCM (Shafiee and Safamehr, 2011) and had harmful impacts on the water resources system of the area, such as reduction of agricultural and industrial water availability and dryness of the Gavkhooni swamp. The Gavkhooni swamp, as one of the most valuable ecosystems in Iran, is an important feature of sustainable development in Central Iran. The swamp provides habitat for over 140 bird species and numerous other flora and fauna. In addition, the Gavkhooni's ecosystem has a vital role in controlling water quality and stabilization of sand dunes located around the wetland. Over the years, population growth and inappropriate water resources management have resulted in a decrease of the quantity and deterioration of the quality of the wetland's incoming fresh water and initiated destruction of this ecosystem (Sarhadi and Soltani, 2013). Some have little hope that this swamp will recover (Madani, 2014; Gohari et.al. 2014). The Eskandari catchment, as one of the most important tributaries of the Zayandehrood River upstream of Zayandehrood's dam basin, plays an important role in the possible restoration and preservation of

the river and Gavkhooni swamp. Inflow fluctuation due to frequent drought periods has recently become an important concern, also in the irrigation sector. The water shortage during dry seasons has intensified the effects of the inflow fluctuation on water users, especially in the downstream section (Hashemy Shahdany and Firoozfar, 2017). The residential areas have spread over the study area from 1997 onwards. Degradation of pasture land and their conversion into residential areas or bare lands over time has increased the surface runoff (Mirzaei et. al. 2016) leading to lower flow in the river during dry periods. The Department of Environment of the Mazandaran province, in the north of Iran where the Kiakola is located, has reported that there are too many wetlands, which have been exposed to drought due to climate change and anthropogenic activities (land use change and overexploiting of (sub-)surface water resources). In the past two decades, the forest area has decreased and the Kiakola catchment has been one of the most affected areas due to rapid urbanization and population growth (in the Mazandaran province, the number of people increased from 960,568 in 1986 to 2,010,948 people in 2011, Statistics Centre of Iran, 2011). This led to higher food and fuel demands, timber smuggling, more intensive livestock grazing, extra road development, exploitation of mines and construction of factories. Changing forest to grassland and overgrazing these grasslands by livestock, has become one of the most major concerns in this area. Grassland were also converted into residential areas. Comparing land use maps from the years 2000 and 2011 showed that rangeland was converted to residential sites (Zare et. al. 2017). A large area of forest land has been turned into gardens and crop land, including rice cultivation (Kavian et. al. 2018). Rice is the second highest consumption product in Iran and the demand is growing each year. Drought in different stages of rice cultivation impacted production (Hasheminya and Dehghannya, 2013).

Please also note the supplement to this comment:
https://www.hydrol-earth-syst-sci-discuss.net/hess-2018-124/hess-2018-124-AC2-supplement.zip

---

## Author Comment (AC3) · 25 Sep 2018

This paper proposes a new methodology to identify positive and negative human modified droughts and tests the methodology on two Iranian catchments. There certainly should be more studies investigating human-modified droughts and it is refreshing to see a case study application with data from catchments in Iran. However, there are a

number of key issues with the study that affect the robustness of the results and conclusions. While I agree with the authors that the methodology could enable quantification of positive and negative human modified drought in the Anthropocene - currently (1) the methods are not very clear which makes it difficult to understand exactly what was done in the study and (2) the methodology only works and attributes droughts correctly if the modelling simulations are robust. The model performance and large differences during the naturalized period do not provide confidence in the validity of the key results of the paper and instead I believe that the human-modified droughts identified are more due to hydrologic model uncertainty rather than human activities. The presentation of the figures also needs to be significantly improved. I encourage the authors to re-address the hydrological modeling they have done, to significantly revise the methods section and to improve the quality of the figures to support their key results.

- The authors appreciate the reviewer's constructive comments. The comments have been incorporated in the revised manuscript. Responses to the specific comments can be found below.

Response:

- In the paper, a step-wise methodology was proposed to make a distinction between different hydrological drought types in the Anthropocene e.g. climate-induced drought, human-induced drought and human-modified drought. We propose a methodology to distinguish and quantify positive and negative human-modified droughts to explore the impact of human interferences on river flow and groundwater. The methodology uses naturalized conditions obtained by simulation modeling as a reference to distinguish the droughts. So as the first step (Step 1), the non-parametric Mann–Kendall test and Pettitt's test were applied to characterize the trends and change point of hydrometeorological variables. Any other technique could also have been used, as long as these distinguish between the natural and disturbed periods. According to the Pettit's test results, the change point divided the study period into two periods: the "natural" and "disturbed" periods. Then at Step 2, hydrological modeling was performed and the

model was calibrated and validated using the data from the natural period. By calibrating the model for the natural period and applying this calibrated model to the disturbed period, the discharge of the disturbed period was naturalized. The HBV light model with the DELAY response routine was selected as a hydrological model (more details about hydrological modeling are presented on P6, section 2.3.2). Clearly any other model could have been used that is able to naturalize disturbed time series. Next, as Step 3, the threshold for the drought analysis was defined for the natural period and applied to the disturbed period. Then, at Step 4, the natural drought (climate-induced drought) and the two human-affected droughts (human-induced drought and human-modified drought) types were distinguished by comparing the naturalized, observed and threshold. Finally, as a final step (Step 5), which is the innovative part of the approach, a distinction between positive and negative human-modified droughts has been made through an extended anomaly analysis. The methodology and each step are broadly described (P3, l7-23). In addition, each step was presented in more detail in Sections 2.2, 2.3, 2.4, and 2.5. We think that these descriptions together with the conceptual diagram (Fig. 3) and the equations in Section 2 adequately explain the proposed methodology. We have moved some text from Section 2 Methodology to Section 3 Results to make the description of the Methodology more clear (Section 3.1, P10 l5-l22 and Section 3.2, P10 l25-l31 and P11 l1-l11).

- The model's performance was evaluated by comparing simulated against observed discharge, especially considering low flow discharge by calculating the Nash-Sutcliff efficiency with logarithmic values of the observed and simulated discharge (lnNSE), which reacts less on peak flows and stronger on low flows than NSE (Krause et. al., 2005) (P7 l9-l21). The NSE$\geq$0.5 has been defined as an acceptable value for model performance (Christiansen, 2012; Moriasi et. al., 2007) (P7 l22-24). The lnNSE for both catchments was equal to 0.5 or higher (Table 1), which we think is acceptable considering the purpose of the study. Although, the model uncertainty will affect the numbers, and other model structures likely will result in other numbers, we believe that the portion of negative and positive human-modified droughts will not substantially

change, e.g. differences similar to what we obtained by using different thresholds (Table 5). Furthermore, we would like to stress that the focus of the paper is on the methodology rather than on the exact numbers for the two Iranian catchments. These are only in the study to demonstrate the methodology (P2, l30-33, P3, l28) Hence, we did not aim at finding a model structure with the best model performance. - In order to support the results of the current research most of the figures were also revised and presented in a better way.

Main Comments

Subdividing the time series into natural and disturbed

Better justification is needed to prove that these statistics are identifying change points. One way could be to gain qualitative data about human activity that is occurring in the catchment (i.e. was a dam built in the change-point year, or a significant new water abstraction implemented?). Figure 1 also needs improving- currently it is really difficult to see any evidence of a distinct change point. As low flows are of interest then discharge could be plotted on a log scale, you should remove the blue dots and I would also add precipitation data. I would also consider plotting a much shorter timeseries (perhaps years that have similar climatic characteristics for both the natural and disturbed period).

Response:

- A justification for change point in discharge time series has been presented in Section 3.1 (P10 l5-l22) for both the Eskandari and Kiakola catchments. Figure 1 (Annex 2) has been deleted and the change point in the discharge and rainfall time series are presented on a y-logarithmic scale (P37, Figure 1 and P38, Figure 2). In addition, the FDC of monthly discharge have been added for the natural and disturbed periods before and after of change point (P 39, Figure 3).

HBV Hydrological Modelling Set Up

a) Objective Function – It isn't clear from the paper whether you used logNSE or NSE. There also needs to be better justification for this use of objective function – there are lots of different metrics that evaluate low flows and you need to better justify your choice here and discuss how it might impact the results.

Response:

- The model performance is evaluated based on the lnNSE, which is appropriate objective function for low flow simulation (P7 l9-l21). It has been frequently used. Each of the statistical criteria has specific pros and cons, which have to be taken into account during model calibration and evaluation. The Nash-Sutcliffe efficiency (NSE) and the coefficient of determination are very sensitive to peak flows, at the expense of being less sensitive to low flow condition. They are based on the squared differences between observed and simulated values (Pushpalatha et. al., 2012; Krause et. al., 2005). Additionally, the coefficient of determination alone should not be used for model evaluation, because it can have still high values for very poor model results, because it is based on the correlation only. In order to reduce the problem of the squared differences and the resulting sensitivity to high extreme values the Nash-Sutcliff efficiency is often calculated with logarithmic values of observed and simulated values (lnNSE), in particular if drought is considered. Through the logarithmic transformation of the discharge values the peaks are flattened and low flows are kept more or less at the same level. As a result the influence of the low flow values is increased in comparison to the peaks values resulting in an increase in sensitivity of lnNSE to systematic model over- or underestimation. So, the lnNSE reacts less on peak flows and stronger on low flows than the NSE (Krause et. al., 2005). So, in the paper the lnNSE has been utilized as an appropriate objective function for low flow simulation.

b) Calibration – The calibration section needs to be much better explained. On Page7, L6 it states that 'The calibration procedure was done more than fifty times' and then on L9 it states 'each calibration was repeated 100'. It isn't particularly clear what the calibration procedure is (simply optimizing logNSE but then R2 is introduced?), how the

parameters ranges were chosen for each of the 14 parameters, how many samples were implemented (50 or 100?) etc. On L9 it also states that this 'resulted in 100 different parameterizations' but it isn't clear how these 100 different parameterizations were used in the rest of the study – I assume you just took the best one to use in the rest of the study but given the uncertainty in the hydrological model, your results would be much more robust if you calculated natural-human drought characteristics for all 100 parameterizations.

Response:

- For the Eskandari and Kiakola catchments the DELAY response routine was used as one of the options in the HBV model, and values of HBV parameters were determined by using the genetic calibration algorithm, which is described by Seibert (2000). Possible ranges of parameter values were defined based on previous studies (Seibert, 1999; Vis et al., 2015). With the genetic calibration algorithm (P6, l26), optimized parameter sets are found by an evolution of parameter sets using selection and recombination. An initial population of n parameter sets is generated randomly in the parameter space and the 'fitness' of each set is evaluated by the value of the defined objective function. From this population a new generation of parameters is produced by n times combining two of the parameter sets. The two sets are chosen randomly, but the chance of being picked is related to the fitness of the parameter set (i.e., the value of the objective function) giving the highest probability to the sets with the highest fitness. In the paper, the calibration procedure was done more than fifty times. Because of the random elements of the Genetic Algorithm and Powell optimization (GAP optimization) used for calibration (Vis et al.,2015), each calibration was repeated 100 times (number of model run for each parameter set), which resulted in 100 different parameterizations. The fitness of each set in the new population is evaluated and the new generation replaces the old one. However, the best set is retained if there is no better set in the proceeding generation. This evolutionary process has been repeated for a number of generations until the maximum number of model runs has been reached (100 times) (P6 l28-l32 and

P7 l1-l8). Finally, the HBV model with the best set of parameters based on the lnNSE was selected for the further analysis. We have decided not to calculate natural-human drought characteristics for all a large set of parameterizations because the focus of the study is on the methodology rather than on the outcome for the two catchments.

c) Calibration Results – It is very difficult to see from Figure 4 the comparison between the observed and simulated discharge. It would be useful to plot a comparison of the flow duration curves (on log scale)

Response:

- Thanks for the suggestion. Figure 4 has been replaced by the FDCs of observed and simulated discharge for the natural period (P29, Figure 4).

HBV Model Results

The attribution of positive and negative human-modified droughts rests entirely on the performance of the model and its ability to represent the naturalised flows. From Figure 5, the observed flows are plotted against the modelled naturalised flows for the period 1987-1988 for Eskandari and 1988 – 1989 for Kiakola. As I understand it from Section 2.2.2 both these periods lie in the 'natural' period (undisturbed period) for each catchment and so you would expect the naturalised model flow to (as much as possible) represent the observed flow and so climate-induced droughts to be identified at the same point in the time series. In this case, there are quite large discrepancies between the observed and modelled naturalised flows and climate-induced droughts are identified at completely different points. This is because the threshold is derived from the observed flows (which are very different to the naturalised flows produced by the model) and so casts doubts on the modelling results as they should be similar for the naturalised period. This has knock-on impacts for the attribution of positive and negative human modified droughts, which again I think are due to model simulation uncertainty rather than human influences. Consequently, it is difficult to have confidence in the results and key conclusions of the paper.

Response:

- Uncertainty is an intrinsic part of the any hydrological modeling. The quality and quantity of hydrometeorological data of arid and semi-arid regions have been the biggest concern of hydrological modeling. In the selected case studies, long time series of observed hydrometeorological data were not available and do not have a high quality. In addition, the spatial distribution of rain gauges in both catchments likely has impact on the hydrological modeling. Besides that, security issues and water conflicts that have occurred over the past years have led to reduction in the availability of information in many parts of Iran. In the paper, we tried to minimize the uncertainty to an accepted level by using a genetic calibration algorithm (GAP optimization) to find an optimal parameter set. Model performance was evaluated by using the lnNSE as an accepted measure by low flow researches. Model performance expressed as lnNSE was not high, but equal or just above the minimum ($\geq$0.5) in both catchments. Hence, we accepted the model for further analysis. We agree that the non-perfect model performance leads to uncertainty in the naturalized flow, which is reflected in differences (i.e. bias) between the observed flow and the naturalized flow for the undisturbed period. Yet, the identification of climate-induced drought is based upon the naturalized flow time series and the threshold derived from these series, which implies that we do not have to account for the bias. The human-modified droughts are more affected, because these are compared against the observed flow. It certainly will have influence on the portions of negative and positive human-modified droughts, similar as the selection of the threshold (Section 5, Table 5). However, the focus of the paper is on the methodology rather than on the actual portions of the negative and positive human-modified drought, which allows us to accept the outcome of the model as a mean to illustrate the potential of the methodology.

References

Christiansen, D.E.: Simulation of daily streamflows at gaged and ungaged locations within the Cedar River Basin, Iowa, using a Precipitation-Runoff Modeling System

model: U.S. Geological Survey Scientific Investigations Report 2012–5213, 20 p, 2012. Krause, P., Boyle, D.P., and Bäse. F.: Comparison of different efficiency criteria for hydrological model assessment, Advances in Geosciences, 5, 89–97, https://doi.org/10.5194/adgeo-5-89-2005, 2005. Moriasi, D.N., Arnold, J.G., Van Liew, M.W., Bingner, R.L., Harmel, R.D., and Veith, T.L.: Model Evaluation Guidelines for Systematic Quantification of Accuracy in Watershed Simulations, Transactions of the ASABE, 50, 885–900, 2007. Pushpalatha, R., Perrin, C., Nicolas, L.M., and Andreassian, V.: A review of efficiency criteria suitable for evaluating low-flow simulations, Journal of Hydrology, 420–421, 171–182, doi:10.1016/j.jhydrol.2011.11.055, 2012. Seibert, J.: Regionalisation of parameters for a conceptual rainfall runoff model, Agricultural and Forest Meteorology, 98–9, 279–293, doi: 10.1016/S0168-1923(99)00105-7, 1999. Seibert, J.: Multi-criteria calibration of a conceptual runoff model using a genetic algorithm Hydrol. Earth Syst. Sci., 4, 215–224, doi: 10.5194/hess-4-215-2000, 2000. Vis, M., Knight, R., Pool, S., Wolfe, W., and Seibert, J.: Model calibration criteria for estimating ecological flow characteristics, Water, 7, 2358-2381; doi: 10.3390/w7052358, 2015.

Please also note the supplement to this comment:
https://www.hydrol-earth-syst-sci-discuss.net/hess-2018-124/hess-2018-124-AC3-supplement.zip